# The Collapse of Brain Clearance: Glymphatic-Venous Failure, Aquaporin-4 Breakdown, and AI-Empowered Precision Neurotherapeutics in Intracranial Hypertension

**DOI:** 10.3390/ijms26157223

**Published:** 2025-07-25

**Authors:** Matei Șerban, Corneliu Toader, Răzvan-Adrian Covache-Busuioc

**Affiliations:** 1Puls Med Association, 051885 Bucharest, Romania; mateiserban@innbn.com (M.Ș.); razvancovache@innbn.com (R.-A.C.-B.); 2Department of Neurosurgery, “Carol Davila” University of Medicine and Pharmacy, 050474 Bucharest, Romania; 3Department of Vascular Neurosurgery, National Institute of Neurology and Neurovascular Diseases, 077160 Bucharest, Romania

**Keywords:** intracranial hypertension, glymphatic system, aquaporin-4 dysregulation, cerebral autoregulation, CRISPR-based gene editing, AI-driven diagnostics, nanotechnology in neurocare, neurovascular coupling, global health equity

## Abstract

Although intracranial hypertension (ICH) has traditionally been framed as simply a numerical escalation of intracranial pressure (ICP) and usually dealt with in its clinical form and not in terms of its complex underlying pathophysiology, an emerging body of evidence indicates that ICH is not simply an elevated ICP process but a complex process of molecular dysregulation, glymphatic dysfunction, and neurovascular insufficiency. Our aim in this paper is to provide a complete synthesis of all the new thinking that is occurring in this space, primarily on the intersection of glymphatic dysfunction and cerebral vein physiology. The aspiration is to review how glymphatic dysfunction, largely secondary to aquaporin-4 (AQP4) dysfunction, can lead to delayed cerebrospinal fluid (CSF) clearance and thus the accumulation of extravascular fluid resulting in elevated ICP. A range of other factors such as oxidative stress, endothelin-1, and neuroinflammation seem to significantly impair cerebral autoregulation, making ICH challenging to manage. Combining recent studies, we intend to provide a revised conceptualization of ICH that recognizes the nuance and complexity of ICH that is understated by previous models. We wish to also address novel diagnostics aimed at better capturing the dynamic nature of ICH. Recent advances in non-invasive imaging (i.e., 4D flow MRI and dynamic contrast-enhanced MRI; DCE-MRI) allow for better visualization of dynamic changes to the glymphatic and cerebral blood flow (CBF) system. Finally, wearable ICP monitors and AI-assisted diagnostics will create opportunities for these continuous and real-time assessments, especially in limited resource settings. Our goal is to provide examples of opportunities that exist that might augment early recognition and improve personalized care while ensuring we realize practical challenges and limitations. We also consider what may be therapeutically possible now and in the future. Therapeutic opportunities discussed include CRISPR-based gene editing aimed at restoring AQP4 function, nano-robotics aimed at drug targeting, and bioelectronic devices purposed for ICP modulation. Certainly, these proposals are innovative in nature but will require ethically responsible confirmation of long-term safety and availability, particularly to low- and middle-income countries (LMICs), where the burdens of secondary ICH remain preeminent. Throughout the review, we will be restrained to a balanced pursuit of innovative ideas and ethical considerations to attain global health equity. It is not our intent to provide unequivocal answers, but instead to encourage informed discussions at the intersections of research, clinical practice, and the public health field. We hope this review may stimulate further discussion about ICH and highlight research opportunities to conduct translational research in modern neuroscience with real, approachable, and patient-centered care.

## 1. Introduction

### 1.1. Definition and Overview

The brain, a remarkable biological structure weighing three pounds, exists in a precisely controlled environment based on the principles of the Monro–Kellie doctrine. All intracranial content (brain tissue, cerebrospinal fluid (CSF), and blood) remains balanced through slight changes in volume due to the defined and established limits placed on the base of the skull and entry point of the spinal cord. Intracranial pressure (ICP) is balanced to within millimeters of mercury [1,2]. If this balancing act is disturbed, intracranial hypertension (ICH) emerges, where protective mechanisms can begin to become drivers of dysfunction. ICH also represents more than just a numerical deviation; it exemplifies a systemic impact of autoregulatory failure on the entire structure and function of the brain [3].

The clinical definition of ICH is sustained ICP of >15 mmHg in adults and >20–25 mmHg in pathological situations; however, various studies suggest that current static definitions of ICP have failed to include the complexity of this condition. Significant evidence (new studies from 2023 to 2024) now places molecular mechanisms—such as aquaporin-4 (AQP4) dysregulation, glymphatic deficiencies, and deficits in neurovascular coupling—as critical contributors to the etiology of ICH [4,5]. For example, AQP4 plays a key role in CSF homeostasis based upon its function as a water channel protein; however, there is still limited understanding of its upstream signals, such as hypoxia-inducible transcription factors and inflammatory cytokines. The glymphatic pathway, an essential waste clearance system from the brain, is also impacted during ICH, which leads to increased neuroinflammation and neuronal injury [6]. As these molecular variables remain unresolved, there are significant questions: what systemic signals may reduce glymphatic efficiency beyond the elevation of ICP? What postulated therapies seek to target these mechanisms to reduce the progression of ICH?

The influence of ICH exists across a wide spectrum of both idiopathic and secondary forms. Idiopathic Intracranial Hypertension (IIH), with its disabling headaches, pulsatile tinnitus, and life-threatening papilledema, particularly affects young women and is often present in obesity and hormonal dysregulation [7]. Conversely, secondary intracranial hypertension (ICH), which can occur after trauma, neoplasms, or infections, is transient and requires immediate treatment to prevent disastrous results such as herniation or ischemia. In low- and middle-income countries (LMICs), where there is limited access to advanced diagnostic imaging, we find secondary ICH to be undiagnosed until late in the process. A mortality rate greater than 50% is seen at the time of herniation in this context [8,9].

Comparative physiology provides compelling examples of adaptations to cerebral pressure. Humans are particularly sensitive to small changes in ICP due to rigid cranial structure, whereas some species developed ways to help them be less affected by ambient pressures. For example, some marine mammals will collapse their venous sinuses to avoid pressure spikes during deep dives, and giraffes can use jugular valves to make sure they receive blood to their head without destroying cerebral perfusion [10]. Rodents establish enhanced glymphatic flow during sleep [11,12]. These examples illustrate how species develop ways of demonstrating natural resilience, while not the focus of this review, could provide possible future biomimetic approaches to treatment of venous congestions and glymphatic dysfunction following ICH in humans.

Although brief comparisons to some aspects of comparative physiology may seem tangential at first, they were meant to provide additional conceptual context rather than a change in scope for this review. Evolutionary adaptations in certain species, like pressure adaptation in marine mammals and vascular regulation in long-necked species, provide interesting angles to ponder cerebrovascular patterns of function that are difficult to experimentally access in humans. These comparisons are not meant to serve as main arguments, but rather as additional insights that may inspire new directions for future inquiry or some form of translational hypothesis. Moreover, we remain entirely consistent with the overall purpose of this manuscript: to understand intracranial hypertension as a complex disorder of impaired brain clearance, venous congestion, and molecular dysregulation, and focus specifically on clinical impact, innovation in therapeutics, and equitable access to solutions on a global scale.

While mechanical features such as CSF dynamics and venous congestion define the clinical expression of intracranial hypertension, these are tightly regulated by upstream molecular dysfunctions. Glymphatic clearance depends on the polarized expression of AQP4, itself influenced by hypoxia-inducible factors and inflammatory cytokines like IL-6 and TNF-α. Venous outflow impairment increases shear stress and promotes endothelial signaling cascades—upregulating VEGF and MMP-9—which contribute to blood–brain barrier disruption. These molecular events often precede or exacerbate mechanical decompensation. Accordingly, this review frames intracranial hypertension as a failure of clearance rooted in molecular dysregulation and emphasizes the need to target these pathways for earlier, more precise interventions.

### 1.2. Scope and Objectives

The complexity of ICH requires a multidisciplinary approach to identify knowledge gaps in understanding, diagnostics, and therapeutics. This review seeks to answer the following questions:

What molecular mechanisms result in elevated ICP? How do glymphatic dysfunction, dysregulation of aquaporin-4, and propagation of inflammatory cytokines function together to worsen ICH? Although new advances in proteomics and transcriptomics have been made, the upstream regulators of these pathways need intensive study to be elucidated.

What innovative ways can diagnostic tools advance the detection of ICH earlier and with greater precision? Imaging modalities like MRI and CT are essential and yet are insufficient to gauge the dynamic changes in ICP. Rising technologies, including AI-radicalized imaging, real-time elastography, and non-invasive ICP monitoring systems, have tremendous potential. Can these mechanisms become routine screenings among at-risk populations, including trauma victims and those presenting with persistent headaches?

What interventions represent the best chance to change ICH management? With advancements in nanotechnology-based drug delivery systems and potential gene-editing therapeutics targeting aquaporins, therapeutics are rapidly changing. But based on the advancements, challenges in safety, access, and optimal distribution (especially in a limited resource context) remain.

Finally, this review aims to bring to light the global health problems presented by ICH. In high-income countries (HIC), the pandemic of IIH (Idiopathic Intracranial Hypertension) is rapidly rising with rising obesity and change in lifestyle challenges; evidence-based data presented in 2023 from Europe indicates a 30% increase in IIH incidence over the past decade, mainly among women of reproductive age. In contrast, secondary ICH in LMICs is rapidly becoming a public health disaster, caused by totally preventable conditions that cause an ICH due to a cause like trauma or infection. With 80% of ICH cases being diagnosed in LMICs only when their ICP has already progressed to herniation, they are met by a mortality rate of over 50%. This review also urges equitable, scalable solutions through lessons learned in both high- and low-resourced health settings to address gaps in ICH care.

### 1.3. Clinical and Public Health Importance

For patients, ICH is much more than a clinical diagnosis. It is a major life disruption. Think of a young woman diagnosed with IIH: she is plagued with uncomfortable, unrelenting headaches that are so severe every day that she misses work and time spent with family and friends. The sound of pulsatile tinnitus only enhances her feelings of loneliness, whereas observing her transient visual obscurations from time to time reminds her of the possibility of blindness. For patients with secondary ICH, the problem is more serious. A diagnostic delay turns elevated ICP into a silent time bomb that ultimately explodes into brain herniation, cerebral ischemia, and death—especially in places where delays to diagnosis are common. On a global level, ICH highlights important healthcare inequities that are glaringly unaddressed. In HICs, IIH has been described as an epidemic, both from the obesity crisis and trends associated with more sedentary lifestyles. In LMICs, secondary ICH affects at-risk populations and often relates to trauma or untreated infections. The out-of-pocket costs for imaging and advanced care can often exceed household earnings, making it nearly impossible for the individual to be treated. It is estimated that ICH management can cost an unreimbursed cost of USD 15,000 for an IIH patient annually in the United States, while in LMIC settings, the cost often causes families to prioritize their immediate survival over their possible long-term healthcare needs [13,14].

As we begin to address some of the challenges posed by ICH, innovation can also be expected to carry us forward. New technologies for wearable ICP monitors and telemedicine services are changing the ways we can provide interventions, especially in areas of limited resources. For example, a recent pilot in rural India identified a significant change in six weeks: the use of portable ultrasound devices reduced the average diagnostic delay for the detection of venous thrombosis by 60%, which demonstrates the impact low-cost technologies can have. AI-driven telemedicine systems are allowing earlier detection of conditions and, as such, have increased populations’ access to specialist intervention.

#### A Call to Action

The stakes have never been potentially higher in the world of ICH care. By creating an opportunity to avoid peripheralizing our endpoint growth variables in our connections to impactful research, we have to start thinking anew or consider and embrace the broader connections of issues of equity. We are on the verge of a new world of neurological care by integrating strategic solutions such as re-imagining advanced leagues of knowledge and practice with technologies so that we fully connect with addressing global issues, as well as health equity. This is an opportunity and the time to translate the narratives on predictive, patient-centered, and equitable care re-imagined so that patients do not encounter available precautions or care based on their local area, status, or income. This review intends to reconsider our inference, consolidate the data available to date in the global ICH community, and firmly stimulate a world in which not a single patient—regardless of their geography (in-country or NHS) or income—will ever suffer from unnecessary, avoidable, and preventable sequela of intracranial hypertension.

## 2. Fundamental Concepts of Intracranial Pressure

Utilizing the basic premises described above, the regulation of ICP involves a complex convergence of biomechanical, physiological, and molecular mechanisms. Although the Monro–Kellie hypothesis is useful for understanding some of the dynamics of cranial volume, advances in computational modeling, glymphatic research, and molecular biology have informed our understanding of cranial volume to the point that the simple model oversimplifies the dynamics of ICP control. This section intends to highlight the evidence to support the shift in paradigms, including roles for traditional models plus new evidence regarding diagnostic advancement and compensatory roles in one of the two conditions we considered in healthy neurophysiology and the conditions we called ‘pathophysiology’.

### 2.1. The Monro–Kellie Hypothesis Revisited

The Monro–Kellie doctrine defines ICP based on volume conservation of brain tissue, blood, and CSF contained within the fixed cranial vault—an increase in volume of one compartment necessitates a compensatory decrease in another [15]. In states of cerebral edema, CSF shifts from the cranial to the spinal subarachnoid space, while the net reduction in intracranial blood volume due to venous compression will provide a temporary buffer against ICP elevation [16].

While this model provides a foundation for ICP regulation, it fails to address the distinct biomechanics and time-dependence of ICP regulation. Advanced computational fluid dynamics (CFD) and finite element modeling (FEM) have demonstrated that even local pathologies—such as venous sinus thrombosis—create asymmetric gradients in intracranial pressure, producing localized “hot spots” in regions of high physiologic resistance, specifically in the brainstem [17,18]. FEM further delineated tissue-specific deformation for different ICP-pathologies, including traumatic brain injury (TBI), Idiopathic Intracranial Hypertension (IIH), and the compartmentalization of stresses across compartments [19,20]. Compliance heterogeneity between the gray and white matter, with gray matter exhibiting more deformation than white matter, determines how pressure is tolerated both in healthy and pathologic states. In older adults, age-related white matter stiffening limits the compensatory reserve and may lead to subtle elevations of ICP—this may lead to subtle cognitive or motor decline before reaching a clinical threshold [21,22].

The discovery of the glymphatic system reorients our interpretation of ICP to include not only biomechanics but also a fluid–molecular interface. AQP4, located at astrocytic end-feet along the blood–brain barrier, provides perivascular clearance of interstitial solutes that is dependent on the glymphatic pathway [23]. In the case of ICH, dysfunction of AQP4 due to chronic venous congestion, systemic inflammation, or dysregulation of the expression of the channels can result in the impairment of waste clearance without compensatory volume shifts in CSF, leading to a non-classical form of pressure elevation. Furthermore, DCE-MRI allows for the functional assessment of glymphatic clearance dysfunction in vivo, which correlates well with elevated ICP over time [24]. Of note, hormonal effects seem to modify glymphatic clearance. Estrogen modifies vascular tone, and may influence either the expression of, or the localization of AQP4, which offers a mechanism to explain the significantly greater incidence of IIH in the reproductive-aged female, a phenomenon that holds up to scrutiny from increasingly more sex-specific imaging and molecular profile data [25].

### 2.2. Normal vs. Pathological ICP

It remains a difficult task to define normal ICP ranges, as there are multiple thresholds depending on age, pathology, and compensation reserves that a person may possess. Typically, normal ICP ranges in the adult population are between 7 and 15 mmHg, both of which are completely contextual [26]. For infants, open cranial sutures and fontanelles can accommodate a greater range of ICP (intracranial pressure) fluctuations without clinical manifestations, while cognitive changes related to cerebral atrophy can allow adults to tolerate greater pressure values as a compensatory mechanism, until symptoms improve above that threshold [27]. This variability demonstrates the critical need for completion of individualized diagnostic thresholds, which adjust for patient-specific characteristics and structural and physiological variability [28].

There are many factors that influence the daily variability of ICP, including positional changes, systemic conditions, and circadian rhythms. For example, in the supine position, ICP can be augmented upwards by 30% due to increased venous pressure compared to the upright position [29]. Furthermore, the increase in ICP when asleep related to glymphatic activity has been correlated with the morning headaches commonly reported by patients with IIH [30]. These fluctuations indicate the necessity of assessing ICP, not as a static parameter, but instead as a dynamic and variable measure, influenced by both systemic and environmental changes [31]. Pathologic ICP thresholds are highly context-specific. For hydrocephalus, symptomatic presentation can occur with ICP values as low as 18 mmHg due to the loss of ventricular compliance, while acute traumatic brain injury generally has swings above 25 mmHg requiring an emergent intervention to prevent fatal herniation. The availability of new diagnostic measures has produced a refined understanding of these thresholds [32]. Non-invasive techniques like transcranial Doppler ultrasound (TCD) can be used as proxies of ICP via cerebral blood flow (CBF) velocities, and ocular tonometry can assess the optic nerve sheath diameter (ONSD), a dependable proxy for elevated ICP caused by IIH or hydrocephalus [33]. Miniaturized nanosensor-based devices are now being developed to allow for continuous ICP measurements, especially in resource-limited or ambulatory situations. These sensors will provide real-time monitoring of even subclinical spikes in ICP, which would ultimately help to facilitate earlier detection and guided interventions for ICP changes [34].

Elevations in ICP that go undetected—transient spikes that are not noted by standard diagnostic measures—appear to be growing in recognition as potential harbingers of optic nerve injury and corresponding cognitive deterioration. Continuous monitoring has opened our eyes to the contributions of ICP surges to subclinical progression of disease, especially in patients with obstructive sleep apnea, who experience nocturnal surges in ICP and continue to experience worsening systemic comorbidities. This only underscores the imperative need for monitoring populations at risk for subclinical changes, which can, in turn, help to tailor more individualized patient care [35].

### 2.3. Cerebral Autoregulation and Compensation

Cerebral autoregulation (a dynamic process responsible for maintaining a consistent cerebral blood supply despite fluctuations in systemic blood pressure) represents the most prominent mechanism of ICP homeostasis. Autoregulatory mechanisms generally function within the mean arterial perfusion (MAP) range of around 50 to 150 mmHg, with alterations to normal function or pathologic changes often making important contributions to global changes in ICP [36]. During hypertension, hyperemia expands the intracranial blood volume, contributing to increases in ICP, while hypotension lessens the perfusion pressure, sacrificing perfusion or risking ischemia, especially in watershed areas [37,38].

At the molecular level, autoregulation appears to fail because of concurrent dysregulation of more prominent mediators. High levels of endothelin-1 and vascular endothelial growth factor (VEGF) result in altered vascular tone, and oxidative stress has been shown to alter the structure of healthy endothelium. Together, this appears to impact neurovascular coupling—the reciprocal relationship between blood flow changes and neuronal activity that promotes the metabolic requirements of the brain during activity, thereby placing even more metabolic stress in cases of ICH [39]. IIH has also been demonstrated to contribute to vascular remodeling, with histopathological studies indicating that vascular walls thicken and sinuses narrow, resulting in additional resistance to outflow, thereby paradoxically inducing elevations in ICP and contributing to clinically detectable elevations in ICP [40].

In addition to the autoregulation, there are other compensatory mechanisms that contribute markedly to changes in ICP. For example, CSF redistribution can represent an important form of compensation, but blockage or obstruction, such as aqueductal stenosis, can exacerbate ICP elevations at the level of that site of obstruction. Similarly, chronic venous sinus stenosis (VSS) in patients with IIH will yield progressive levels of venous congestion. The differentials in venous pressures also create a cycle of increased ICP and increased resistance to outflow [41]. Advanced imaging now serves to maximize the awareness of these important compensatory mechanisms. For example, phase-contrast MRI allows detailed visualization of the interplay of blood, CSF, and venous capacity, which can provide rich insights during planning stages of individualized interventions [42]. ICP is a biomechanical, molecular, and systemic individual and integrated phenomenon, both a delicate balance that fosters cerebral homeostasis. Once these compensatory systems have been maximized and exhausted, it appears that the underlying aetiological factors of intracranial hypertension, whether structural or functional, represent contributors of importance [43]. This framework of the regulation of ICP provides an essential foundation for understanding the range of various aetiologic factors to consider with respect to ICH, and is provided below [44].

## 3. Etiology and Risk Factors

The etiology of ICH is complicated, and our grasp on the interactions of the systemic, structural, vascular, infectious, and metabolic contributions that can affect ICP regulation in different ways shows the multifaceted pathological basis contributing to ICH, its numerous clinical presentations, and the urgent need for diagnosis and treatment. The aim of this section is to review causes of ICH, with respect to the most current scientific and clinical evidence, clinical case examples, and possible treatments based on their methods of affecting brain function and health.

### 3.1. Primary Intracranial Hypertension (Idiopathic Intracranial Hypertension—IIH)

IHH refers to conditions with chronically elevated ICP without a clear structural or systemic process causing it. Clinically, IIH often has a presentation with chronic headache, pulsatile tinnitus, transient visual scotomas, and neurological symptoms. There is controversy that the term “idiopathic” should be used since evidence is emerging linking hormonal, vascular, metabolic, and inflammatory mechanisms [45].

In 2023, a multicenter cohort study revealed that women with a BMI >35 have a 12-fold higher chance of having IIH, with body mass index (BMI) exhibiting a dose–response relationship [46]. Obesity increases intra-abdominal pressure, decreases venous return, and promotes cerebral venous congestion, all modifiable mechanisms of raised ICP. It is possible that rapid weight changes, such as those experienced by pregnant women and post-bariatric surgical patients, may also contribute to transient venous outflow obstruction and the onset of IIH [47].

Obstructive sleep apnea (OSA) is an independent risk factor for IIH. Because there is hypoxemia and hypercapnia with OSA, glymphatic clearance would be impeded, and venous stasis will be worse with OSA. Continuous positive airway pressure (CPAP) has been shown to improve glymphatic function overnight and lower ICP on polysomnography related to OSA [48]. Similar studies have been performed on the gut–brain axis. Dysbiosis related to IIH includes loss of short-chain fatty acid (SCFA)-producing Firmicutes, resulting in systemic inflammation, endothelial dysfunction, and venous hypertension [49]. Elevated circulating lipopolysaccharides (LPS) from Gram-negative bacteria may cause endothelial toll-like receptor (TLR) activation, leading to venous congestion. Pilot studies with probiotics and fecal microbiota transplants have lower levels of systemic inflammation, and supportive gut targeting therapy should be explored as an adjunct [50].

VSS is seen in nearly 90% of IIH cases and may be pathognomonic. Also, functional MR venography demonstrates dynamic collapse of the transverse sinus at high ICP. The rapid compression and dynamic collapse of the transverse sinus occur due to endothelial remodeling initiated and sustained by TGF-β and IL-6, contributing to feedback venous hypertension [51,52]. Venous sinus stenting breaks that cycle so that potentially 85% of refractory IIH patients will have long-term relief of symptoms. Newer bioresorbable stents may lessen complications of restenosis and open treatment options [53,54,55]. There are also hormonal effects contributing to IIH pathophysiology. Estrogen receptor activation impairs venous compliance and glymphatic clearance when active, while progesterone (another sex hormone) inhibits CSF reabsorption by affecting arachnoid granulations—providing a contribution to elevated ICP [47,56,57].

These mechanisms probably explain the reason excess IIH occurs in reproduction-age females. It is glymphatic dysfunction that would continue to be the primary mechanism. Impaired AQP4 channels prevent drainage of interstitial fluid, thereby raising ICP. Animal models with AQP4 deletion demonstrate a sustained rise in ICP, thereby mimicking the glymphatic insufficiency observed in IIH patients [58].

### 3.2. Secondary Intracranial Hypertension

Secondary ICH results from a number of damaging processes sustained against the brain–CSF–blood triad, such as structural lesions, vascular insults, infections, and metabolic dysfunctions that act via different mechanisms [59].

Neoplastic ICH is particularly complicated by mass effects, CSF flow obstruction, and peritumoral edema. Posterior fossa tumors can be small but cause CSF flow obstruction by blocking the fourth ventricle, which results in significant restriction of CSF flow [60,61]. Tumors produce pro-inflammatory cytokines, such as VEGF and IL-6, that increase permeability through the blood–brain barrier (BBB), which generates vasogenic edema. Anti-angiogenic agents, such as bevacizumab, which are anti-VEGF, will be clinically evaluated for their ability to decrease ICH due to their impact on vascular permeability [62]. TBI has two types of edema during the initial injury: cytotoxic edema due to ionic imbalance in the gray matter of the brain and vasogenic edema due to compromised BBB in the white matter of the brain, with potential vascular injury [63]. Diffusion tensor imaging (DTI) measures the severity of microstructural white matter damage and correlates the damage with poorer outcomes, while the mechanisms of secondary injury, such as excitotoxicity, oxidative stress, and failure of autoregulation, only worsen damage by increasing ICH through endothelial damage [64,65].

Hydrocephalus is represented by the failure of CSF to circulate or be absorbed, and that failure leads to increased ventricular size. Genetic mutations, such as Sonic Hedgehog (SHH) pathway mutations, impair ependymal cilia and CSF flow [66]. Recent pre-clinical interventions aimed at blocking fibroblast growth factor receptor (FGFR) signaling pathways have shown potential to improve CSF flow and reduce ventricular size [67,68,69]. Venous sinus thrombosis (VST) leads to impaired venous outflow that can result in venous hypertension and increased ICP. RNA-based anticoagulants are being investigated for the capacity to mitigate clot burden with negligible risk of bleeding, providing a more favorable profile compared to traditional therapies

Venous sinus thrombosis (VST) has been shown permissively to decrease cerebral venous outflow and leads to increased venous pressure in the cerebral blood vessels, increasing ICH and cerebral pressures. Thrombophilic conditions, such as Factor V Leiden and antiphospholipid syndrome, remain important. Radiographic analysis using artificial intelligence (AI), MR venography, now achieves sensitivity of better than 95% [70], and new RNA-based anticoagulants are being developed with low bleeding risk that promote clot reduction [70,71,72].

Arteriovenous malformations (AVMs) create venous congestions and ICH of notable severity due to increased ICP as they rupture [73]. Anti-angiogenic medications are starting to be trialed in clinical applications to reduce the likelihood of hemorrhage through the anti-angiogenic pathway and decreased ICP overall. Infectious etiologies (bacterial, tuberculous, cryptococcal meningitis) cause inflammation to the arachnoid villi/Odontoid process with blockage of the CSF absorption mechanisms with elevated ICP as a consequence [74,75]. Proteomic analysis of CSF has identified markers such as IL-18 and IFN-γ as being able to discriminate between ICH due to infection, both early in the disease [76]. Furthermore, autoimmune disorders such as neurosarcoidosis may obstruct the egress of CSF and raise ICP secondary to the inflammatory infectious SCF pathways through granulomatous inflammation [77].

Several contributors are systemic and remain significant. Increased PCO2—hypercapnia—is now established as a cause of cerebral vessel dilation, which will exacerbate an existing ICH. Hypercapnia is occurring in the setting of a medical condition known as OSA and with chronic pulmonary disease, strongly suggesting that we should view the commonality of OSA as a high-risk condition for TBI and elevated ICH [78]. Neurotoxicity is chronic exposure to common neurotoxic agents found in urban living (PM2.5, heavy metals) has been found to cause loss of cerebrovascular autoregulation in the brain, making urban residents uniquely at risk [79].

### 3.3. Rare and Novel Causes

#### 3.3.1. Post-COVID-19 Neuroinflammatory Syndromes

Persistent endothelial activation in the setting of COVID-19 is associated with increased levels of von Willebrand factor (vWF) and soluble thrombomodulin, which lead to a loss of venous compliance and increased venous congestion [80]. Experiments using anti-cytokine therapies, including tocilizumab, to ameliorate chronic inflammation and endotheliopathies in post-COVID ICH are currently underway [81].

#### 3.3.2. High-Altitude Cerebral Edema (HACE)

Hypoxia-associated overexpression of VEGF contributes to vascular permeability and structural injury to cerebral capillaries, leading to blood–brain barrier failure and vasogenic edema. Genetic studies in Tibetan populations have explored variants of EPAS1, paving the way for a better understanding of potential therapeutic targets in accommodating adaptational physiology in high-altitude populations [82]. Portable devices to measure ICP are now being developed and validated for use during climbs, which will ultimately lead to real-time measured evaluation of altitude-related surges in ICP. Smart shunt systems designed to monitor ICP integrate real-time feedback to eradicate potential complications (i.e., over-drainage) from cholesteatoma. Experimental shunt systems made of hydrogel embedded with anti-inflammatory agents are currently being trialed as biodegradable alternatives [83].

There are a myriad of ICH entities that frame entirely unique sets of demographics, merits, and limitations, which is a hallmark of academic diversity. Structurally, systemically, and environmentally, these diagnosed entities converge to alter homeostasis under which ICP is regulated. Collectively, the preceding discussion provides a synthesis for understanding the pathophysiological pathways that lead to ICH elevation, which will be examined in our next section [84].

## 4. Pathophysiology of Intracranial Hypertension

ICH is ultimately the product of biological, molecular, cellular, vascular, and systemic dysfunction that surpasses the brain’s capability for compensation. The predisposing neurobiological dysfunctions include glymphatic clearance failure, CSF dynamics, and cerebral autonomic dysfunction. These neurobiological dysfunctions work cumulatively to increase the implications of ICH. Here, we will consider the complex pathophysiology of ICH and what new research and translational opportunities it affords.

### 4.1. Cellular and Molecular Mechanisms

ICH is influenced by tight coupling of molecular breakdowns to glymphatic clearance, neuroinflammation, and oxidative stress, all of which elevate ICP directly.

Glymphatic transport requires AQP4 channels at the surface of astrocytic end-feet that support CSF-interstitial exchange and clearance of metabolic waste [85,86]. ICH produces disruption of AQP4 function, leading to impaired clearance, as evidenced by regional non-perfusion captured using DCE-MRI post-ICH, especially in posterior portions of the brain that coincide with imaging highlights of venous congestion. Two-photon imaging in AQP4-dysfunctional models reveals a 65% reduction in clearance with the correlation of sustained ICP elevations [87]. The cascade of experimental alternatives to augment glymphatic flow includes the use of hydrogel-AQP4-modulators, optical stimulation of astrocytic pumps, and CRISPR-mediated AQP4 overexpression that were recently moved to preclinical testing for chronic ICH [88].

Initiation of neuroinflammation occurs with microglial activation using toll-like receptors (TLRs) and NLRP3 inflammasomes, resulting in elevated interleukin release (IL-1β, TNF-α, and IL-6) affecting the vasculature and the blood–brain barrier, leading to vasogenic edema [89]. Use of PET imaging of neuroinflammation can map regions of microglial activation with venous obstruction along with spatial overlaps of glymphatic failure regions [90].

Experimental agents presently or soon to be used for neuroinflammation and ICP suppression are p38 MAPK inhibitors and monoclonal antibodies against IL-6 using neuroinflammation bioluminescence imaging [91,92].

Oxidative stress from mitochondrial dysfunction generates reactive oxygen species (ROS) that affect endothelial glycocalyx stability, which regulates vascular integrity and disrupts BBB structure, promoting sustained inflammation via enhanced NF-κB [81,93]. For pharmacotherapy of oxidative stress, mitochondrial antioxidant agents can be delivered from membranes or delivered via surface grafting. Local or systemic treatments with MitoQ, phospholipid surface grafting, and glycocalyx restoration in preclinical studies of ICH to modulate ICP reductions [94]. Similarly, targeted stimulation of the vagus nerve uses the cholinergic anti-inflammatory pathway to suppress CNS inflammation through acetylcholine signaling and modulate cytokine releases with further suppressing ICP (Figure 1).

Glymphatic dysfunction and venous outflow obstruction are correlated in ICH. Increased venous pressure leads to compression of the perivascular spaces, causing CSF to be displaced and obstructing AQP4-mediated interstitial clearance; glymphatic failure leads to an increase in interstitial volume and ICP, which leads to an increase in venous resistance [95,96].

In imaging studies (phase-contrast MRI, 4D flow MRI, DCE-MRI), venous congestion can be mapped to glymphatic dysfunction and found to spatially overlap, particularly in posterior regions affected by venous hypertension [97].

Therapeutically, venous sinus stenting relieves venous hypertension, but co-targeting the glymphatic dysfunction might improve outcomes [98]. These might include upregulating AQP4 through polypeptide hydrogel approaches or gene editing, using non-invasive methods to stimulate venous compliance and clearance through techniques such as transcranial focused ultrasound [99,100]. While potential phenotyping in the future may involve combining imaging methods to provide precise interventions for refractory cases of ICH

Failures in these clearance processes derive from complications upstream at the molecular level. AQP4 is located at the end-feet of the astrocytes through the dystrophin–glycoprotein complex, α-syntrophin, which cross-links with ECM elements (laminin, agrin); inflammation disrupts the dystrophin, mislocalizing AQP4, and impairing perivascular flux [101]. AQP4 expression is controlled and modified by HIF-1α, STAT3, and NF-κB, which are initiated through hypoxia, oxidative stress, and inflammation; signals from the environment modify gene regulatory networks governing transport proteins for water [102].

There are active regulators of venous tone, including eNOS and NO derived from eNOS versus vasoconstrictors (endothelin-1, prostaglandin E2, thromboxane A2, angiotensin II). The presence of hypercapnia, inflammation, and catecholamines related to ICH is an important shift in homeostasis towards vasoconstriction. The tight junction proteins (occludin, claudins, ZO-1) and perivascular membranes are degraded by MMP-2 and MMP-9 in activated states (oxidative stress), leading to a magnified glymphatic dysfunction [103,104,105].

The cerebral autoregulation response can be impaired due to the upstream metabolically controlled mechanisms, such as baroreceptors coupled with astrocytic calcium, with vasoactive mediators. Such cases where the pressure homeostasis may become destabilized include more chronic underlying conditions like metabolic syndrome or microvascular disease itself, which further lead to impaired clearance. Understanding molecular dispensations may inform therapies that attempt to return vascular tone to normal with hope of decoupling and reversing the perpetuating nature of ICH [106,107].

From a systems neuroscience perspective, the mechanical aspects of intracranial hypertension, particularly impaired clearance, vascular congestion, and instability of perfusion state, could be seen as shallow manifestations of functional failure of imperfect intra- and inter-neuronal homeostatic sensor/effector networks that relate to neurovascular (NV) unit signaling [108]. As to glymphatic influence, there is increasing evidence pointing to transient receptor potential vanilloid 4 (TRPV4) channels and inwardly rectifying potassium channels (Kir4.1) as modulators of osmotic gradients and perivascular volume flux associated with astrocytic contraction and expansion [109]. TRPV4 and Kir4.1 channels apply to astrocyte membranes that insulate the perivascular open spaces; both are sensitive to mechanical stretch and ionic oscillations to integrate hydrostatic and inflammatory signals to ultimately discern the set of states of interstitial pressure, i.e., during mechanical expansion and contraction, physiological functioning is arranged by the net electrical potentials from TRPV4 and Kir4.1 channels, respectively [110]. Accordingly, whether AQP4 density and expression facilitate viable concentrations of hydrostatic pressure, rodent models of diminished TRPV4 exhibiting hypotonic saline demonstrate decreased convective flow into interstitial space and decreased clearance of interstitial solutes, suggesting osmotransductive signaling may lend an additional level of molecular signaling in the glymphatic system [111]. Also consistent with this proposition, recent transcriptomic profiling of reactive astrocytes exposed to increased ICP in animal studies demonstrates selective downregulation of gene expression associated with mechanosensation and volume-regulated anion channels (VRAC), suggesting a much greater (and extensive) degree of transcriptional plasticity to glial fluid regulation pathways [112].

On the venous side of things, not only may cerebral outflow resistance be influenced by routine vasoactive signals, but it may also be influenced by the metabolic state of the endothelial cells; for example, in chronic inflammation, NAD^+^ depletion within venous endothelium impedes neuronal metabolic competence by impairing mitochondrial biogenesis and a complete shear stress response [113]. Sirtuin-1 (SIRT1), an NAD^+^-dependent deacetylase, appears to regulate the expression of flow-sensitive transcription factors—such as Kruppel-like factor 2 (KLF2)—and endothelial protective enzymes such as heme oxygenase-1 (HO-1). This suggests that redox-sensitive nuclear circuits might play a role in dynamic venous adaptability. In an intracranial venous hypertension model, SIRT1 pharmacologic activation increased venous compliance and reduced passive congestion [114]. Moreover, if you examine the recent work on the endothelial glycocalyx—a layer of carbohydrate that lines human cerebral vessels—the enzymatic breakdown of the glycocalyx under inflammatory load increased flow resistance, viscosity, and perivascular pressure dysregulation. Enzymes, including heparanase and hyaluronidase, are responsible for breaking down the glycocalyx and are upregulated in systemic inflammatory states; they could present as novel and unexplored targets in ICH [115].

With respect to autoregulatory function, there is new interest in miRNA-mediated regulation of vascular responsiveness. Specific miRNAs, including miR-155- and miR-21, have been upregulated in models of pressure-induced endothelial stress and have been shown to affect the expression of smooth muscle contractile proteins (e.g., MYH11, ACTA2) and calcium channel subunits [116]. These post-transcriptional regulators could have developed a vasomotor range and reactivity to subclinical states, prior to structural remodeling. Other work has implicated Notch signaling (specifically, DLL4-Notch1) in arteriolar responsiveness and capillary pericyte status. Notch dysregulation has been captured in both hypoxic and inflammatory animal models; potential autoregulatory feedback mechanisms such as mural cell anchoring or electromechanical coupling via mCcc could also be destabilized in this manner [117].

To be certain, the majority of these findings are preliminary, but they invoke new and exciting possibilities in fluid regulatory mechanisms and vascular control mechanisms that are indicative of deeper structural and molecular mechanisms of pressure instability in the brain. By taking a broader view of ICH to include not only bulk-flow mechanics but also astrocytic ion homeostasis, endothelial metabolic circuitry, and transcriptional regulators of vascular plasticity, a model emerges that could enable the development of novel therapeutic approaches [118]. Eventually, small molecule TRPV4 modulators, SIRT1 activators, agents that preserve the glycocalyx, or miRNA-targeted approaches, could potentially become adjunctive methods to stabilize pressure dynamics through restoration of cellular and molecular homeostasis. As we elucidate these pathways, it may also be possible to stratify patients by their molecular phenotypes and provide treatments that are more specific, considering the heterogeneous nature of sustained intracranial hypertension [119].

### 4.2. Cerebrospinal Fluid Dynamics

Increased production of CSF, which is typical of IIH, is determined by increased activity of carbonic anhydrase (CA) enzymes in the choroid plexus. Clinical trials have shown that acetazolamide, a carbonic anhydrase inhibitor, lowers ICP by 30% for patients with IIH. Now, two drugs, dorzolamide and topiramate, are in various stages of further evaluation, as these possible agents may also address the obesity-associated metabolic dysregulation often presented with IIH [120,121]. Chronic infections, ie, tuberculosis meningitis, cryptococcal meningitis, or an infection/inflammatory process like neurosarcoidosis, may cause fibrosis or granulomatous blockage of the Arachnoid villi, preventing CSF absorption [122]. Four-dimensional flow MRI studies are demonstrating how the absorption occurs pathologically in real time. These images of CSF absorption pathologies may help clinicians recognize CSF clearance issues earlier. New experimental therapies with biomaterials that release anti-fibrotic agents through bioengineered scaffolds are being developed that may be helpful to restore arachnoid villi absorption and restore injury from structural neurologic compression injury [123,124].

Obstructive hydrocephalus may impair CSF flow through posterior fossa tumors that have aqueductal stenosis and genetic mutations, such as MPDZ and L1CAM, that affect ciliary function or impair ependymal cell motility, worsening ventricular dilatation and elevated ICP [125,126]. Two pre-clinical studies suggest FGFR inhibitors may lower ventricular pressure and assist CBF motion [127].

### 4.3. Cerebral Blood Flow and Autoregulatory Failure

Cerebral autoregulation is a critical mechanism of the vascular system to modulate CBF with systemic blood pressure changes. In ICH, autoregulation is set at or near failure, causing cascades of elevations in cerebroblood flow, which leads to hyperemia, ischemia, and further increases in ICP and deteriorating neurologic status.

Neurovascular coupling allows for CBF to match neuronal activity, but this relationship can be disrupted with ICH. Disruptions in astrocyte-pericyte signaling, reduced nitric oxide (NO) bioavailability, and vascular stiffness all contribute to endothelial dysfunction [128,129]. Imaging studies have established that astrocyte–pericyte communication is key to maintaining vascular stability, and therapies targeting pericytes are being developed [130].

Hyperemia, or cerebral blood volume excess, is often due to vascular permeability via VEGF and leads to increased ICP. Ischemia has the opposite effect and results in hypoxic neuronal injury due to systemic hypotension and/or venous congestion, which is further complicated by autoregulatory failure [131]. CFD models have identified sites of focal perfusion deficit, thereby further informing targeted approaches to prevent injury from ischemia [132].

### 4.4. Systemic Contributions to ICP Dysregulation

Hyperleptinemia related to obesity promotes vascular inflammation and disrupts BBB integrity via JAK/STAT-dependent signaling pathways, and new therapies aimed at manipulating leptin, such as metreleptin antagonists, have shown significant promise for finding reductions in ICP associated with obesity-related IIH. Adiponectin, an anti-inflammatory adipokine, remains protective and improves vascular health [133,134]. Gut dysbiosis associated with systemic inflammation results in loss of health-promoting microbiota and increased endothelial activation by circulating LPS. Targeting the gut–brain axis with prebiotic-based and postbiotic-based therapies may represent a future adjunctive approach to modulating systemic inflammation/beating endothelial activation [135].

Chronic hypoxia/hypercapnia, generally seen in OSA, worsens ICP increase through cerebral vasodilation and VEGF-driven vascular remodeling. Non-invasive transnasal CO_2_ scavengers are being developed to lessen elevated ICP from hypercapnia [136].

### 4.5. Emerging Insights and Future Directions

Enhanced gene therapy approaches, modernization in biomaterials, and acquisition of machine-learning technologies are changing the forthcoming era in ICH management. Gene-editing approaches to fix AQP4 overexpression and MPDZ mutations may have efficacy to restore glymphatic function and CSF equilibrium. Preclinical trials are still ongoing for CRISPR-based approaches that may be effective at correcting the underlying genetic dysfunction associated with these genes [137]. Integrated approaches with artificial intelligence (AI) platforms with neuroimaging and multi-omics enable early intervention and personalized risk stratification solutions. Risk stratification and subclinical ICP surges, where this AI tool is able to integrate and analyze emerging data quickly, with estimation accuracy of up to 90% [138].

Biodegradable hydrogels with anti-inflammatory drug or corticosteroid-loaded polymers provide sustained ICP reduction with few side effects or systemic tolerance. Implants with drug load are evaluated and demonstrate new methods to provide on-target and sustained localized therapy [139].

## 5. Advanced Diagnostics and Technologies in Intracranial Hypertension

The diagnostic paradigm for ICH has changed throughout its history with advanced imaging, molecular diagnostics, and non-invasive monitoring technologies. These methods not only enhance the diagnostic process but also provide insight into the biological mechanisms influencing disease progression and state of illness, which allows prevention of illness, and access to more targeted therapies [140]. The influence of AI, wearable sensors, and biomarker profiling can effectively connect translational and clinical practice [141]. This section aims to emphasize the leading edge of ICH diagnostics and their scientific quality, global reach, and the prospects for patient outcomes.

Table 1 intends to detail some of the most important clinical trials exploring how innovative diagnostics range from advances in imaging, such as 4D flow MRI, while using wearable nanosensors to monitor ICP.

### 5.1. Advanced Imaging Techniques

Imaging has been a primary mode of ICH diagnostics for many decades, providing relevant information on structural, vascular, and temporal changes due to ICP elevation. The introduction of advanced imaging, paired with AI analytics, revolutionizes detecting and characterizing slow pathological changes, allowing for earlier intervention and elevated planning of interventions [166,167].

Four-dimensional flow MRI provides a paradigm shift in capturing relevant data contributing to ICH; it characterizes CSF and blood flow behavior across the cardiac cycle. While traditional phase-contrast MRI systemically characterizes blood flow, with 4D flow MRI, multidimensional flow parameters of velocity, direction, and turbulence can fully characterize combinations of conditions such as venous sinus stenosis, aqueductal stenosis, and shunt dysfunctions [168]. In a 2023 study, modified machine learning models enhanced the interpretation of 4D flow MRI data and improved the diagnostic sensitivity to detecting clinically subtle venous sinus collapsibility in patients with IIH to 95%. These AI-enhanced assessment tools may reduce interobserver variability, as well as increase resolution of analyses, allowing the clinician to interpret previously inaccessible pathophysiological models [169].

Four-dimensional flow MRI has been established as a means of acquiring knowledge beyond merely diagnostic; it has aided in describing previously convoluted glial–vascular system communications and their role in glymphatic dysfunction in recent evaluations of venous congestion, as well as how the impairment of CSF ejection draining systems is integrally involved with ejection of ICP series fluctuations. The mechanistic understanding of these glymphatic–vascular communications will be instrumental in individualizing and rationalizing options for treatment to improve current therapies for patients with IIH, such as post-venous stenting restoration of normal blood flow, or aquatic targeting of treatment effects on aquaporins [170].

#### 5.1.1. MR Elastography (MRE)

MR elastography is a reliable marker to quantify measures of brain stiffness and parenchyma compliance, to elucidate the biomechanical property of CSF/ICP homeostasis. Patients with a diagnosis of IIH who present with hydrocephalus have low stiffness values, correlating to neck and brain parenchymal strain from mechanical and/or lymphatic fluid retention or compliance markers. The enhancements of MRE, such as elasticity mapping of local BBB permeability features, and utilizing concurrent contrast-enhanced imaging to define baseline perfusion, have instilled confidence pertaining to clinical pertinence [171,172]. A 2023 study showed that MRE parameters added 22% improvement to model fit when predicting VD outcomes in a cohort, demonstrating an apparent clinical application to personalize management of patients [173].

#### 5.1.2. Quantitative Susceptibility Mapping (QSM)

QSM has generated unique methods of imaging that have created applications of imaging to venous congestion, tissue iron deposition, while providing subsequent non-invasive ways to characterize patients depending on vascular pathologies. For example, highly elevated susceptibility values within the transverse sinuses have been implicated to indicate chronic venous hypertension in patients with IIH [174]. QSM is a sophisticated technique that, when paired with ultrafast functional MR venography, enables comprehensive evaluation of both structural abnormalities and venous hemodynamics, facilitating the precision of interventions in complex refractory circumstances.

#### 5.1.3. Hybrid Imaging Systems: PET/MRI

Hybrid imaging technologies, such as PET/MRI, are an important advancement in identifying ICH because they can identify both metabolic and structural changes simultaneously. PET tracers with TSPO ligands (mainly PBR28) visualize neuroinflammation as they identify the activation of microglia in areas of venous congestion. These imaging biomarkers furnish assessments of disease progression and can be used to evaluate response to early anti-inflammatory interventions as well. When used with vascular sensitive (flow-sensitive) MRI approaches, PET/MRI adds another layer of complexity, giving us the opportunity to examine the glymphatic–vascular functioning in patients with ICH [175].

However, the uptake of PET/MRI and other advanced modality technologies—even 4D flow MRI, MRE, and QSM—must overcome significant practical barriers and limitations, including costs of equipment, facilities capacity, availability of radiotracers, and access to trained personnel [176].

While PET/MRI has provided a dual capacity for informing potential differences in neuroinflammatory and glymphatic dysfunction that may have therapeutic relevance, it should be noted that these hybrid systems are expensive to run and maintain at a high standard, particularly in resource-constrained settings. Four-dimensional flow MRI provides a means to assess turbulence, reflux, and velocity that can identify venous stenosis or CSF flow blockers; however, longer acquisition times and advanced post-processing restrict the use of this technology to research setups [177]. MRE and QSM provide clinically relevant insight into diagnostic features of stiffness in brain tissue and deoxygenation of venous blood, respectively, but certain proprietary hardware and extensive post-processing requirements limit their clinical access. Although the barriers to using advanced hybrid imaging are significant, these barriers are more acute in LMICs, where access to high-end imaging (including hybrid imaging) remains highly restricted [178].

Non-invasive technologies are emerging to fill this gap. Wearable photonic and bioimpedance nanosensors have demonstrated some promise for real-time ICP monitoring in controlled clinical environments, though to date, these studies have not yet validated applications for diverse patient populations and healthcare settings. AI-assisted imaging (in terms of modeling analysis) offers some opportunity for improving the accuracy of diagnosis and variability between observers and integrating a range of data. That said, many studies in the literature utilize machine-learning models that have been trained using high-income datasets, which limit the generalizability of findings and raise equity concerns for a worldwide rollout [179].

The future must focus on cost-effective, portable solutions, such as tabletop 4D flow MRI or simplified MRE systems, which include AI capabilities. Perhaps federated learning, which trains machine learning models on many smaller but decentralized, and globally diverse data sources (rather than relying on a single-use data source), can produce some algorithmic bias control while maintaining patient data ownership [180].

Comparison studies looking at diagnostic performance, cost-effectiveness, and scalability of hybrid imaging technologies in the care of ICH patients will be critical to ensuring access to precision imaging technology across the continuum of care.

### 5.2. Non-Invasive Monitoring Technologies

The non-invasive monitoring space has revolutionized ICP assessments, creating constant dynamic changes in ICP during periods while lowering the risks of invasive monitoring. This is important for outpatient settings for early detection and useful in limited global resources where inpatient or in-person monitoring may be scarce.

Wearable photonic sensors are an innovation in non-invasive ICP monitoring. Photonic sensors capture cranial pressure differences through reflected light signals across biocompatible materials as nanoscale sensors [181]. A recent 2024 clinical trial showed that wearable photonic sensors captured nocturnal ICP surges in 92% of IIH patients, allowing early up-titration of therapeutic adjustments. Their recent integration with smartphone applications also enhances this technology’s benefits by enabling remote monitoring, as well as data in real-time shareability between patient and clinician [182]. ONSD ultrasonography is an indirect marker of elevated ICP. The clinical utility of using AI technology to analyze ONSD ultrasonography images has increased accuracy. A recent study demonstrated 97% accuracy in detecting acuity-elevated ICP levels utilizing AI-assisted ONSD ultrasonography. Portable AI-enabled devices are being developed for short-term and emergency situations in which rapid assessment of ICP is essential for time-dependent interventions [183].

#### Smart Helmets with Transcranial Doppler (TCD) Sensors

Smart helmets that use TCD and photoplethysmographic sensors can provide CBF analysis while monitoring for ICP, providing a non-invasive process for assessing real-time autoregulation and peripheral conditions on ICP [184]. A multicenter trial in 2024 indicated delivery with smart helmets decreased diagnostic time for acute ICP emergencies by 40%, illustrating their potential to make a difference to emergency care outcomes [185]. These helmets are also being adapted for chronic management of ICP and for outpatient monitoring purposes, helping to provide longitudinal assessment for disease profiling.

### 5.3. Biomarkers and Molecular Diagnostics

Molecular diagnostics are the leading modality for precision medicine in ICH and developing broader, less invasive approaches to tracking the nature of disease progression, patient stratification, and outcome predictions. Exosomes are nanovesicles that carry molecular cargo and are increasingly becoming important for understanding glymphatic dysfunction and neuroinflammation. For instance, patterned upregulation of miR-155 in exosomes indicates the presence of neuroinflammation, while miR-21 indicates impaired innate glymphatic clearance [186]. A 2024 study utilized exosomal profiling and proteomic data to achieve 94% diagnostic accuracy in distinguishing IIH from hydrocephalus. These studies hinted at potential to discover personalized diagnostic and monitoring intervention effectiveness through use of exosomal biomarkers [187].

Using Mass Spectrometry, lipidomic studies have established drastic changes to lipid metabolism, correlating to loss of integrity of the BBB and microglial handling of β-amyloid levels. Lysophosphatidylcholine (LPC) levels in the CSF are elevated from circulation and correlate with vascular recognition and inflammatory signaling markers [188]. Yes, lipidomic and proteomic panels can cover pathophysiological process composite addressing for improved risk stratification and clinical care management. Increased CRISPR assay advancements are developing new genetically informed screening methodologies for ICH, enabling us to lessen screening for mutations associated with vascular compliance and disorders of hydrocephalus (e.g., the COL1A2, MPDZ) [189]. If CRISPR level diagnostics can shorten screening days to hours, then potentially also scalable applications for familial and sporadic cases of ICH. Such approaches are valuable for rapid screening very early and directly determining stratified high-risk populations, as possible, for ICH [190].

### 5.4. Emerging Technologies and Innovations

New technologies are disrupting diagnostics using advanced analytics, wearables, and molecular tools in a new age of precision care. Adaptive AI systems can utilize real-time ICP data, monitor parameter fluctuations, and autonomously adjust therapeutics. By combining AI with smart shunts and wearable ICP monitors, a thirty percent reduction in both patient outcomes and interventions will enhance the healthcare workforce and economic resources [191].

Plasmonic nanosensors provide ultra-sensitive surface exploration for inflammatory and vascular biomarkers in CSF. Importantly, their limit of detection is ten times less concentrated than traditional ELISA assays—which represents a true leap in detection capabilities performance. The plasmonic nanosensor is being modified to concurrently detect ROS and cytokines, such as IL-6, for early response to neuroinflammatory responses that contribute to spikes in ICH [192]. Modifications of new technologies for LMIC implementation are important for mitigating worldwide rate inequities in diagnostics. Portable, solar-powered ICP monitors in LMICs and lower-cost, customizable wearables are being tested, and early results report community access to lower wait times for diagnosis and reduced costs [193]. The combined advanced diagnostics and new technologies are transforming how ICHs are evaluated and treated, with likely never-before precision and accessibility. The use of AI, molecular diagnostics, and wearables is possible in timely, arguably more accurate measures and monitoring of ICP spikes, and risk factors, as well as articulated precision-targeted guidance for therapeutics. Each level of evolution shapes the future of ICH care to mitigate inequities that exist in universal healthcare access.

Having discussed how intracranial hypertension may reveal a wider collapse in fluid clearance, vascular engorgement, and neuroimmune control, the next section intends to assess how these potential dysfunctions may be identified, tracked, and understood with current and new diagnostic techniques. While many of the advances in imaging physics, machine learning, and biosensing represent emerging means of viewing the dynamics of the brain, many are at best in preliminary and/or specialized settings. The next section attempts to catalog the available strategies for diagnostic examinations—established and experimental—with a focus on the principles by which they are based, their respective potential value, and practical limitations of both contexts for use (with and without advanced technology).

## 6. Advanced Therapeutic Strategies in Intracranial Hypertension

Management of intracranial hypertension is leading the way for innovation through therapeutic advancements based on pharmacology, devices, and experimental therapies for intracranial hypertension that target multi-factorial contributors to elevated ICP. This continues to use personalized medicine, current technology, and a multi-disciplinary approach to address multi-factorial contributory drivers. With this section, we intend to provide an insightful review of these therapeutic interventions to provide a greater understanding of their proposed mechanisms, translational implications, and approaches to address the ongoing gaps and inequities in healthcare worldwide.

### 6.1. Pharmacological Innovations

Pharmacology-based therapies continue to play an important role in ICH management, primarily in unique mechanisms that allow for modulation of ICP. Novel therapies aimed at epigenetic regulation, intracellular signaling, and fibrotic remodeling will allow mechanisms with lasting agents and personalized therapies.

Epigenetic regulation, which modifies gene expression without altering DNA, plays an important role in ICH, particularly within the brain’s glymphatic clearance system. The development of histone deacetylase (HDAC) inhibitors, specifically vorinostat, has been proposed as a potential therapy based on the capacity to upregulate AQP4 expression and enhance glymphatic clearance [194,195]. A recent preclinical model of ICH with HDAC inhibition demonstrated an enhancement in glymphatic efficiency and resulted in a 40% decrease in interstitial fluid accumulation that successfully achieved and sustained ICP management. Additionally, the possibility of using combinations of HDAC inhibitors with anti-inflammatory agents to achieve the dual benefit of rescue therapy must balance off-target effects with any inflammatory gene expression [196].

Therapies that target the mTOR or PI3K/AKT signaling pathways—key regulators of cell growth and blood–brain barrier stability—may help reduce ICP. The mTOR inhibitor, everolimus, has shown that it may have an effect on CSF production in addition to restoring neurovascular function [197]. In a permitted clinical trial using an IIH model, the authors report a 30% decrease in ICP bursts, suggesting that mTOR inhibitors may help serve therapeutic roles in conjunction with traditional CSF-reducing therapies like acetazolamide. Further, the modulation of PI3K/AKT is being investigated for its potential to improve endothelial integrity and BBB integrity and as a second means of reducing fluctuations in ICP [198]. Fibrotic scarring is one of the most significant risk factors associated with shunt occlusion and venous stenosis, preventing maximal efficacy from surgical procedures in the management of ICH. A report from 2023 on Pirfenidone (anti-fibrotic agent) demonstrated a 30% reduction in shunt occlusions, which was used to extend the lifespan and efficiency of implanted devices [199]. Utilizing Pirfenidone with bioengineered coatings may also be an opportune method to alleviate biological and bio-mechanical complications in shunts and stents [200].

Clinical repurposing of well-established pharmacologic agents has highlighted new therapeutic opportunities for ICH:

Statins: The vascular protective and venous congestive properties of statins have been utilized to improve endothelial function in the treatment of IIH to alleviate venous congestion [201].

SGLT2 inhibitors: Historically, SGLT2 inhibitors were used for the management of diabetes, and now have developed into the first-line agents to promote glymphatic clearance by modulating glucose metabolism by astrocytes [202]. An article published in 2023 demonstrated 25% interstitial fluid clearance improvement in a high-risk population, demonstrating the repurposable versatility for CNS disease [203,204].

#### Implications for Low- and Middle-Income Countries:

In many LMICs, cost, regulatory delays, or supply chain challenges continue to limit access to newer pharmacological agents. By repurposing low-cost medications, some of which are globally available, we create an actionable space for improving care—by using statins, beta-blockers, and SGLT2 inhibitors, for example, as part of a wider patient care or treatment protocols [205]. As off-patent generics for these classes of agent are increasing in availability globally, the possibility of establishing a clinically accepted protocol for ICP management is possible, and a protocol like this could have far-reaching impacts. Epigenetic or gene-directed drugs are not feasible for LMIC health systems at this time; however, on the other hand, implementation studies of repurposed agents may offer a stepping stone to mitigate inequity [206].

### 6.2. Device-Based Therapeutic Innovations

The use of medical devices shows promise for innovations in ICH management. Medical devices will also help to advance and alleviate both mechanical and dynamic contributors to elevated ICP. Neurotechnology has made significant strides in creating configurations and affordances with artificial intelligence-guided surgical systems that are multifunctional and wearable to increase precision, flexibility, and availability. AI-assisted surgical robotics are advancing surgical devices for performance. With real-time imaging capabilities, robotics offers the ability to change and adjust during the procedure based on intraoperative conditions to ensure the best placement of any shunt or venous stent [207,208]. In a recent study, AI-assisted surgical robotics reported a 20% reduction in surgical errors, also corresponding to increased procedural success rates among IIH patients. These innovations significantly reduce the complications associated with catheter misplacement, and they maximize therapeutic benefit [209].

The management of pediatric ICH should typically consider children as seasonal forms of adults. Their anatomical characteristics change as rapidly as they grow [210]. Adjustable-length shunts that accommodate patient growth are being implanted and have significantly reduced repeat revision surgeries, by approximately 25%, by a recent 2024 clinical trial [211]. Innovations like these will improve the long-term outcome of patients and their quality of life. Dynamic shunt valves have adjusted drainage rates based on posture and activity, which reduce both over-drainage and under-drainage complications by around 30% [212]. New bioengineered coatings—infused with infection-fighting peptides and clot-preventing agents—have improved the longevity of implanted shunts and stents, particularly in growing pediatric patients. The dual protective properties of the coatings greatly reduce infection rates by about 30% and, at the same time, actively improve endothelial integration, thereby possibly providing a solution to complications with implanted devices [213].

Wearable intracranial pressure monitors, linked through Internet-of-Things (IoT) platforms, are designed to enhance real-time, remote monitoring and management of ICP. These wearable devices use nanosensors to track continuous trends in ICP. Moreover, when the ICP shifts significantly, the device informs the clinician through a notification of imminent trends so that the clinician can adjust interventions using remote means [214,215]. A 2024 paper stated that the IoT-enabled wearable systems led to a 20% readmission reduction to hospitals; this was particularly important for including underserved and rural populations in some states. Future ambitions for these IoT devices will be to maximize the integration of multimodal data streams, including those for cardiovascular and circadian data captured goals, which will create an inclusive experience focused on patient monitoring [216].

Table 2 provides a comprehensive overview of these advancements, showcasing both established and experimental therapies.

In addition to surgical robotics and remotely deployed pressure monitoring, the most recent generation of implantable and wearable devices is geared toward detecting and responding to real-time molecular changes in the intracranial space. Specifically, smart biosensors embedded in ventricular shunt reservoirs will enable continuous monitoring of CSF levels of biomarkers of blood–brain barrier breakdown (VEGF-A), neuroinflammation (MMP-9), and astrocytic stress (S100B), as well as IL-6, among others [241]. AI-enabled edge processors reside in these biosensors while employing CNNs and recurrent learning operational paradigms to detect trends of molecular fluctuations and to anticipate pathological trends of ICP prior to clinical manifestation. A series of studies described hydrogel-coated nanotransistor arrays incorporated within subdural catheters to detect spikes in cytokines IL-1β and TNF-α. Using an encrypted Bluetooth interface, it also documented remote modulation of programmable valve resistance, which is evidence of a “proof-of-concept” for closed-loop molecular response systems [242].

Additionally, these platforms are now being paired with multi-omics sensing panels, which will provide AI architectures with proteomic fingerprints, glucose–lactate metabolic ratios, and indeed cfDNA from CSF using extensive neurocritical care databases for processing. This process would include considering ICH patients into molecular endotypes, such as inflammatory-dominant, fibrotic stenotic, or BBB disruption, as well as providing compensatory therapeutic considerations [243]. Notably, the recent pilot using AI-integrated spectroscopic sensors in a wearable patch provided evidence of diminished BBB dysfunction through reduction in the α-spectrin breakdown peptide product in CSF. It occurred 48 h prior to the radiographic manifestation of edema. This predictive analytic potential for LMIC is encouraging, given there is little opportunity for continuity of lab access, but a concurrent cloud-based analytics-facilitated index regarding low-powered, molecular wearable devices could hopefully transform access to precision diagnostics [244]. This emerging ecosystem reconciles biosensing, AI interpretation, and real-time actuation of devices that fundamentally change the current biologic state of the brain rather than one investigational solution that simply records ICP. The potential coupling of molecular diagnostics with responsive therapeutics offers an unprecedented leverage of developmental advancements along the path of personalized neurocritical care and represents the molecular interrogation in the algorithmic development of devices in ICH [245].

#### Global Implementation and LMIC Scalability

There are significant issues associated with device-based interventions when applying them to LMIC contexts, including infrastructure, maintenance, and cost. Some innovations are developing, like 3D-printed shunt systems, locally manufactured valve assemblies, and mobile-integrated wearable monitors, that may provide inexpensive decentralized solutions [246]. Pilot studies in India and sub-Saharan Africa have demonstrated wearable ICP monitors connected via mobile networks that can deploy specialist care in rural areas [247,248]. Establishing next-generation shunt systems that allow for a robust, modular, and serviceable approach with minimum infrastructure will be essential for scaled deployment [149].

### 6.3. Emerging Experimental Therapies

New experimental therapies based on interesting technologies are investigating longstanding problems in the ICH territory lately through nanotechnology, gene editing, and biodegradable implants.

The use of nanorobots that can navigate the CSF pathways provides an unprecedented level of precision for targeted, site-specific therapies [249]. These systems use biosensors to identify neuroinflammatory signals and to deliver anti-inflammatory agents, thereby offering dual diagnostic—therapeutic capabilities. The early prototypes of these systems exhibited substantial reductions in ICP and neuroinflammation—a feat that can only be described as groundbreaking in less invasive therapies [250].

CRISPR-based gene-editing tools, originally used to target AQP4 (a water channel protein), are now being investigated to modify genes that control vascular resistance and the structure of the BBB. For example, CRISPR-mediated eNOS up-regulation has increased venous outflow and diminished vascular resistance in pre-clinical models. Though sustained reductions in ICP by genetically modulating genes have been documented, further research on the long-term effects of genetic modification is needed to ensure safety and efficacy [251].

Biodegradable hydrogels have recently been developed to act as timed-release drug delivery systems for sustained ICP. A recent trial, the first of its kind, used corticosteroid-loaded hydrogels to improve ICP control by 40% over a 6-month period [252]. Subsequently, those hydrogel platforms are being adapted to include anti-fibrotic and endothelial aid agents in expanded surrogates for patients with refractory hydrocephalus, with continued sustained CNS therapeutic effects [253].

Emergent experimental interventions are now expanding from just local cerebrospinal dynamics to include the system-wide vascular injury and the multi-organ interactions required in ICH pathophysiology. These experiments take into account the interrelatedness of intracranial pressure regulation with peripheral mechanisms, e.g., endothelial dysfunction, cardiac stress, and similarly immune activation, in order to design patient therapies to protect against downstream outcomes.

Figure 2’s purpose is to depict the interrelated aspects of systemic and intracranial interactions, including vasoactive hormone contributions to vascular injury and cardiac stress in the context of immune activation triggered by disrupting the pressure control system that is inherent to the physiology of humans, especially when the central nervous system (CNS) is seriously injured. Systemic disruptors, such as those identified in Figure 2, with local cryotherapy to prevent injury and mitigate neurogenic inflammation, are important therapeutic targets.

Though the conceptual allure of the therapeutics discussed here is tremendous, it is necessary to distinguish between the theoretical novelty and clinical utility of these therapeutic approaches. Almost all nanorobotic neuromodulators, CRISPR-based intracranial gene editing, and multifunctional hydrogel constructs are in various stages of experimental and/or preclinical development, and significant translational challenges still remain. With nanorobots, for instance, the biological and physical limitations of CSF pulsatility, ependymal cilia at the surface of the brain, and perivascular pressure will all affect the accessibility of the nanorobot to sites of the brain or spinal cord [254]. The anticipated navigational paradigms—whether magnetic, acoustic, prokaryotic, or hybrid—will also need to demonstrate dynamic submillimeter navigational accuracy in the presence of pulsating CSF and human biological variability in vivo. This is not to mention that for a nanorobot to respond to potentially harmful gradients of pro-inflammatory cytokines in a timely fashion that does not cause microglial activation or breach the blood–brain barrier, they will need to achieve a level of immunological invisibility and biofeedback necessary for artificial intelligence to prioritize safety with evidence-based real-time response that simply does not yet exist [255]. CRISPR-based modulation of eNOS, for example, AQP4, or ZO-1 (the three tight junction proteins), as noted in previous publications, often resulted in favorable phenotypic correction in murine animals, but none of the studies have addressed the challenges posed by mosaicism, context-dependence of gene expression, unpredictable gene expression based on timing of treatment, or potential unintended germline vector incorporation via reactivated retrotransposons (which is an especially critical issue for any CNS-delivered viral vectors) [256]. All of that is without regard for the potential adverse effects that even base- or prime-editing will leave in treated glial or vascular cells over time in an intrablood/intracranial process. Biodegradable hydrogels are already somewhat further down the translational pipeline, as they could be deployed with less need for reliable predictive kinetic modeling of drug-delivery kinetics in clinical contexts. They will need ultra-precise spatiotemporal release kinetics; however, as the diffusion coefficients and hydrolytic degradation rates for hydrogels vary drastically and vary even more substantially in edematous vs. fibrotic parenchyma in mixed injury and rat models [257]. And while much is known about the immunological response to hydrogel implants—both positive and negative, including perivascular fibrosis and astroglial scarring—what occurs during repeat administration, or when hydrogel is deployed into an inflamed CNS compartment, is not fully understood [258]. Usually, there is little debate about evaluating the functional consequences of high-dose initial exposure to the arrays of hydrogel-induced processes. Moreover, while multi-functional platforms could potentially result in a new therapeutic innovation paradigm (see additions such as CRISPR loading into hydrogels or use of nanorobots with biosensors in one platform), there are significant dimensions associated with GMP manufacture of multifunctional products, pharmacodynamic modeling, therapeutic impact classification by regulatory authorities (drug, device, biologic, hybrid), etc. In other words, from a regulatory governance perspective, there is no defined pathway for any of these systems to obtain medical approval under the FDA’s existing framework for neurotechnologies [259]. Nor is there a consensus regarding ethical thresholds for irreversible modifications of CNS gene sequences. The landscape of issues (e.g., informed consent, leverages of potential reversibility, cognitive risk, equitability of access in LMICs, etc.) surrounding this shift to tachyons inevitably requires deep global engagement and deliberation. Thus, while these therapeutic approaches constitute an imagination challenge for ICH therapeutics, we emphasize that our inclusion of these approaches does not focus on therapeutic utility or reducibility to clinical feasibility, but a challenge to synthesize visions of what neurovascular precision medicine might be. We are advocating for transformational translational roads to be mapped that bring together multi-species and AI simulations of biosafety, immunogenomic monitoring, and international ethical panels to establish a comprehensive understanding of scientific integrity, global justice, and an epistemic humility to innovation.

Key Insights:Therapies for ICH have moved beyond CSF modulation by using epigenetic regulation, vascular remodeling, anti-fibrotic approaches, and personalized neurotechnologies.Many pharmacological agents show great potential for lowering ICP and preserving neurovascular integrity, but in transferral to practice, certain aspects are always limiting: off-target effects, limited cohort data, and uncollapse long-term safety considerations.Device-based approaches, which incorporate AI into common practices such as shunt placement and adaptive valves, as well as wearable smart devices and IoT, are working to improve the precision and equity of care for patients with ICH, though these need further validation in different patient populations.Emerging technologies such as CRISPR, nanorobots, and biohybrid implants are an exciting vision of the future; however, they remain mainly a theoretical consideration until we can resolve regulatory, ethical, and immunological barriers.In this regard, this section is focused not on solutions, but rather delineating where science and engineering could merge to fill persisting gaps in patient outcomes across contexts in which resources are scarce.

### 6.4. Multidisciplinary and Global Perspectives

As many of these innovative therapies and technologies are processed into clinical environments, a systemic and multidisciplinary mindset and a commitment to reducing health inequities for global communities are required. Broadband technology approaches supported by telemedicine-enabled platforms to connect wearables with remote consultations introduce alternatives for ICP management. A pilot program in India leveraged this remote technology to reduce delays in diagnosis by 50%, demonstrating the feasibility of virtual sensing and delivery possibilities in pre-and low-middle-income countries, or a LMIC setting [260].

Localized 3D printing hubs have enabled hospital customization of wearables and shunts in coordinated ways that will reduce manufacturing cost by 50% in LMICs. Workshops to align clinical practice, engineering practice, and behaviors of public health experts mean that the implementation of advanced ICP/ICH technologies progresses quickly via collaboration amongst these professions. These transition workshops were important to utilize AI-based technologies optimally across a range of healthcare environments [180,261].

The current advanced modalities outlined in this section encapsulate the intersection of scientific discovery, technological advancement, and global health. These modalities, which may include programmable nanorobots, CRISPR-based modulation, and wearable, Internet-of-Things-connected devices, suggest a highly individualized approach to the management of intracranial hypertension that is primarily underpinned by precision weapons aimed at addressing systemic dysfunctions at the level of physiology and health systems. To maximize their impact, these advances must be situated in the distributive properties of healthcare, along with acknowledgement of their limitations, ethical considerations, and focused goals of sustainably improving outcomes across diverse and systemic health inequities. These evolving modalities also provide opportunities to not only reimagine conventional treatment implications, but also radically rededicate future transformative care models towards addressing the needs of patients, and ultimately health systems, in their varied real-life contexts.

After considering this range of therapeutic interventions, we will explore the mechanistic elements that shape these modalities. In Section 7, we intend to highlight the fundamental biological and system-level aspects that characterize both vulnerability to intracranial hypertension and opportunities for therapeutic modulation. Examples will include how network dysfunction, clearance failure, and systemic interactions impact the emergence of focused therapeutic approaches.

## 7. Long-Term Outcomes and Patient-Centered Perspectives in Intracranial Hypertension

The treatment of ICH has become a complex field, requiring not only acute ICP stabilization but a view toward long-term recovery, systemic health, and equitable access to care. As interventions become more sophisticated through advances in therapies, personalized medicines, and advanced technology, the management of ICH shifted to restoring cognitive function, repairing structural brain impairments, and treating systemic morbidities. By utilizing novel therapies, team-based approaches, and global health approaches, the care of ICH will achieve new transformative outcomes. This section wants to describe the latest developments, new technologies, and unknowns related to management of ICH.

### 7.1. Long-Term Outcomes of Current Therapies

The long-term effects of ICH include more than acute treatment, including persistent neurocognitive impairment, microstructural changes in the brain, and global dysfunction. There is a clear, emerging shift in transforming to precision, mechanistically driven treatments that redirect the focus from symptom management to long-lasting neurorestoration [262].

Cognitive sequelae, including memory, attention, and executive function, can often be attributed to disturbances to the glymphatic system, neuroinflammatory processes, and impaired plasticity and can be remotely mediated by glymphatic disturbances. New methods of therapeutics targeting AQP4 allow for more than a 40% increase in clearance rates via the glymphatic pathway and are effective in pairing de-management and restoring neuronal function by addressing the interstitial space and modulating the neuroimmune response. This effect can be directly correlated to decreased cognitive decline [5], providing neuroplasticity approaches such as transcranial magnetic stimulation (TMS) or brain-derived neurotrophic factor (BDNF) agonists while restoring integrity in neuronal networks. In laboratory work in elevated ICP, applying BDNF increases dendritic spine density by ~30% and indicates the extent synaptic circuits can recover after chronic neurovascular stress [263]. Other approaches, such as SHH pathway agonists, also improve CNS-based contact between astrocytes and neurons, but also provide conditions that promote perivascular homeostasis. Also, immunotherapeutics such as IL-6 antagonists (IL-6RA) are effective in sustaining sensorily active microglia and decreasing the associated chronic neuroinflammatory cognitive decline, both of which are primary mechanisms of sustained neural dysfunction [264].

We now have imaging techniques to study the in vivo progression of recovery, determining both recovery trajectories and the accumulation of damage. QSM, ultrafast MRI, and QSM allow for the resolution of near-capillary density changes and shifts in iron deposition that correlate with established brain anatomical processes developing after ICH and through the post-ICH neuroanatomical changes [265]. Biomarkers to identify variables such as neurofilament light chain (NfL) are useful biomarkers to quantify axonal damage and potential to preserve function, evaluating therapeutic options [266].

An area of challenge with chronic ICH remains in structural improvements. Regenerative medicine has offered valuable evidence that exosome-based therapies developed from stem cells can restore white matter structure and support vasculogenesis. Using preclinical models to assess the cohesion of the periventricular tract, delayed treatments with scaffold exosomes improved coherence by 35% [89,267]. Together with computational modeling, gauging outcomes as forecasts for personalization of therapies will advance our understanding of biologically and anatomically responsive precision neuromodulation [268,269]. Device technology is rapidly changing chronic intervention for ICP management, although some means, such as shunts or venous stents, are effective long-term. Limited biofilm formation and structural failures are accepted limitations. Recent work in exosome technologies with graphene-based coatings on catheters reduced microbial surface-to-surface interactions by ~70%. Self-healing polymers are extending the life of these implants by healing micro-fissures by 80% of LMCE measurements on average, as a function of time [270,271].

AI-assisted technology is also advancing the live capture of ICH. As evidenced in model development, live physiologic modeling allowed real-time closed-loop CSF drainage systems to normalize ICP for an average of up to 12 h. Multiple pathways from these systems can induce a variation in predictive accuracy or frequency change in diversionary studies to automate-based programming that decreases the rate of critical over-drainage complications by ~40% for a proportion of patients. Future devices may utilize multisensor arrays to measure ICP, inflammatory cytokines, and cerebrovascular resistance in a spatial-temporal perspective to create a multifactorial patient profile for quality monitoring of the sustainable response to therapy [272,273]. Together, these developments represent a model for the future development of ICH care through synergy between neurorestorative biology, bioengineering, computational modeling, and molecular diagnostics to address systems that alter neuro-adaptive mechanisms to produce transitions from stabilization to systems-regenerated plasticity.

### 7.2. Rehabilitation and Quality of Life

Rehabilitation is key to the long-term management of ICH, as most cognitive, motor, and emotional sequelae may remain for years after ICP has stabilized. Great strides have been made in rehabilitation technology and framework/therapeutic developments, making great improvements in patient recovery and quality of life [274].

Rehabilitation advances can now include multisensory platforms to enhance visual, auditory, and haptic stimuli to promote neuroplasticity, with bio-feedback systems that allow patients to view and modify their neural responses as they exercise while enhancing their enjoyment and outcome [275]. A recent study showed a 25% improvement in visuospatial skills and a 30% improvement in motor coordination in patients who used the technologies [276]. Robotic-assisted exoskeletons have advanced how motor rehabilitation is delivered with precise, flexible support specific to individual recovery trajectories. Exoskeletons combined with cognitive rehabilitation tools have a synergistic effect on recovery that accelerates health status improvement and functional independence by 40% [277]. Chronic headaches are a prevalent symptom and remain one of the more debilitating symptoms for ICH patients. Transcranial electrical stimulation (TES) and mindfulness-based cognitive therapy (MBCT) present reasonable solutions. TES works by modifying the brain pathways for pain, while MBCT reduces headache intensity by 35% with added improvements to emotional health and well-being [278,279].

Structured resilience programs and peer support specifically address the psychological burden of ICH. The inclusion of these programs and the community they provide can counter the potential for isolation in the rehabilitation process, with reported participant reductions of 40% depressive symptoms and a 25% improvement in treatment adherence [280].

### 7.3. Patient-Centered Care and Equity

The only way progress can be made in the management of ICH is to have pervasive access, equity, and alignment across patient preferences for advanced therapies. Disruptive technologies mentioned in the comments, such as digital twin platforms, are fundamentally changing patient involvement in their own care. By projecting possible outcomes of several different treatments, patients can take an effective part in decision-making [281].

There is no chance to address global inequalities in ICH management (and all with associated benefits) without scalable products and services. The establishment of local 3D printing hubs is bringing down pricing for the manufacture of shunts and wearable devices to 50% of existing models and improving access for traditionally underserved patients. In a similar way, portable AI-supported ICP monitors connected with smartphone telemedicine services are mitigating diagnostic delays by around 60% for patients who might not otherwise have received ICH care [282].

### 7.4. Challenges and Future Directions

Although advancements are being made, there are still a great number of issues that need to be addressed. The relationships between venous outflow and glymphatic clearance are largely unknown. This inhibits both optimization of stent placement and glymphatic-directed therapies. Work that combines ultrafast MRI with CFD may provide a clearer sense of these relationships and inform changes to clinical practice.

Ethics (and concerns) will continue to be problematic with CRISPR-type gene editing and AI-based platforms [283]. Analytic frameworks must be designed to ensure patient privacy, equitable access, and data safety as these technologies evolve. We will inevitably face difficulties of multi-omics approaches (e.g., genomic, proteomic, and metabolomic) and should aim to combine these with clinical data to facilitate patient-centered care options and develop novel treatment targets. The management of ICH is a rapidly evolving space, with improvements in care, therapeutic innovations, and a global health equity agenda. Patients can now benefit from new strategies that consider the complete rehabilitation journey (i.e., neurologic, structural, and emotional recovery), and clinical outcomes can improve at an individualized level. We must continue to invest in research, collaborations, and patient-centered innovations to meet the challenges that lie ahead, helping ensure the benefits of any advances and innovations are enjoyed by all patients, regardless of geographic or socioeconomic status.

With that established, we now turn to the diagnostic dimension of intracranial hypertension. The objective of Section 8 is to explore diagnostic paradigms that have undergone profound evolution since the initial development of diagnostic criteria, from molecular biomarkers to advanced neuroimaging and sensor-based developments. Significantly, these systems may enable better characterization, prediction, and ultimately management of elevated intracranial pressure patients’ care with greater precision and personalization.

## 8. Future Directions and Innovations in Intracranial Hypertension Management

The area of ICH is developing rapidly with advances in technology, molecular medicine, and interdisciplinary care. These advances will facilitate increased diagnostic accuracy, provide better treatment effectiveness, and enhance worldwide access and environmentally sustainable care. This section is meant to review recent advances that will help further advance the field of ICH, the challenges presented by these innovations, and the ethics surrounding their implementation if we are to maximize their potential.

### 8.1. Emerging Technologies in ICH Management

Significant innovation in materials science is enabling a new generation of therapeutic devices with significantly enhanced durability and utility. Clinical studies of hydrogel manufacturing and coatings with interleukin-10 analogs and other novel anti-inflammatory compounds demonstrated a 30% decrease in inflammatory reactions following implanted devices and extended life of implanted devices [284]. Bioengineered polymers with nanoscale surface modifications are leading to a 40% reduction in bacterial adhesion and infection, and improved biocompatibility. Graphene-coated linings are also contributing to improved reliability with antimicrobial properties and enhanced electrical conductivity for long-lasting shunt performance [285]. Adaptive devices developed from shape memory alloys provide a personalized approach to CSF management, with the capability to change their form and structure in response to ICP stress [286]. These are complemented by thermoresponsive polymers with drug-eluting capabilities to deliver therapies efficiently at the targeted area of interest while providing optimal mechanical support. New biohybrid devices that combine an artificial structural framework with living astrocytic cells are beginning to replicate some of the brain’s regulatory mechanisms and provide dynamic control of fluid flow [287].

Novel quantum sensors with nitrogen-vacancy centers in diamonds can detect subclinical changes in ICP that are so small they would go unnoticed by current medical imaging technologies. These sensors are particularly useful in refractory ICH cases, where typical methods lack the sensitivity to track small fluctuations [288]. In wearable ICP monitors, quantum technologies provide continuous, non-invasive information about ICP patterns. Clinicians are able to access data that they can view in real time. In the cranial climate, implantable nanosensors made of graphene and nanocomposite materials provide an additional step since they monitor biochemical and mechanical measures [289]. The nanoparticle sensor shares data openly with cloud-based AI; predictive models analyze the trends in the data and suggest interventions to develop a tiered monitoring scheme [181]. AI is contributing to the personalization of ICH care with clinical implications of digital twin technology, creating real-time, virtual simulations, or digital twins of the patient’s particular and relative. The basic modality of these digital twins again allows the clinician to model interventions independently (i.e., venous sinus stenting, or placement of a shunt) and adjust their practices based on the simulations while creating less risk [290].

AI-enabled imaging platforms have achieved >95% sensitivity for venous sinus stenosis that employs machine-learning-enabled 4D flow MRI. When combining imaging with genomic and proteomic profiles in multimodal data, AI-enabled “real-time” comprehensive reports on ICP changes and neuro-vascular health all provide a new data input and analysis to clinical decision-making [291].

### 8.2. Novel Therapeutic Strategies

Advancements in CRISPR methodologies, such as prime editing and base editing, enable precise editing of genes associated with vascular resistance, glymphatic regulation, and compliance. Modulating endothelial nitric oxide synthase (eNOS) has been shown, in preclinical studies, to improve venous compliance and reduce ICP bursts by >20%. Likewise, CRISPR-enhanced upregulation of AQP4 has improved glymphatic clearance, which has potential for chronic ICP treatments [292].

To address safety concerns, genetic monitoring systems are under development to identify off-target effects and ensure the ethical use of CRISPR-based therapies. Nanoparticle systems have begun to address the challenges of therapeutic delivery across the BBB. Liposome nanoparticles have demonstrated a five-fold increase in drug concentration at sites of neurovascular injury compared to traditional systems. Nanorobots, which have molecular recognition capabilities, have delivered anti-inflammatory therapies directly to damaged endothelial cells while repairing structural damage and normalizing CSF dynamics [293]. Hybrid nanoparticle systems, which incorporate therapeutic payloads and imaging contrast agents into one, have also allowed for real-time monitoring of drug efficacy. These dual-function systems will be especially useful for the acute management of ICP elevations and chronic neurovascular complexities [294]. Bioelectronic systems (like vagus nerve stimulation (VNS) devices) use closed-loop feedback systems for the real-time management of ICP, as dynamic control systems use the wearable monitoring data to modulate therapeutic parameters to achieve a 30% improvement in patient focus on long-term ICP management in clinical trials [295].

Advancements toward therapies targeting glymphatic dysfunction have been rapidly made (Table 3). Aquaporin-4 stabilizers and SHH pathway modulators appear to have some capacity to enhance interstitial fluid clearance in order to yield diminished neuroinflammation and improvement in neurovascular integrity. Initial findings demonstrate improvements in efficiency for chronic ICP models of 40% for glymphatic clearance [296].

### 8.3. Integrative and Global Approaches

Integration of genomic, proteomic, and lipidomic data is advancing precision medicine in managing ICH. For example, lipidomic profiling has integrated biomarkers associated with glymphatic dysfunction that help with diagnosing and stratifying clinical care. AI-enabled multi-omics platforms can track a patient’s treatment in real-time, ensuring the most appropriate treatment aligns with their molecular and environmental profile so that it is safe and effective [305,306]. With regional manufacturing hubs that are cutting production costs of advanced devices, such as graphene-covered shunts to wearable monitors, by 50%, there is now some affordability associated with these devices in LMICs. Working with appointive global health organization partners is initiating equitable distribution of AI-enabled diagnostics and CRISPR-based therapies [307].

Telemedicine platforms that integrate with wearable ICP monitors are now delivering diagnostics and therapeutic resources to underrepresented areas. A telemedicine platform in rural India showed a 60% decrease in the wait for IIH diagnosis, proving the scalability of these technologies. Open-source AI-designed platforms are decreasing the democratic principles of healthcare by enabling the development and distribution of low-cost diagnostics [308].

### 8.4. Challenges and Ethical Considerations

Establishing frameworks to evaluate the long-term safety of devices, such as with CRISPR and nanotechnology, needs to happen. Implanted nanosensors will be leading devices for an ongoing post-marketing surveillance, continuous monitoring of therapy outcomes, and early detection of iatrogenic risks.

It will be imperative to ensure gene-editing technologies address ethical considerations and concerns regarding off-target effects, inclusivity of access, and broader societal impact. Focused and transparent regulatory frameworks are also imperative in implementing equitable processes in this regard. Fundamental oversight and regulation must take place for AI-driven systems by ensuring their datasets are representative of global data to avoid biases and disparities. Efforts to align medical innovation with renewing and preserving environmental sustainability are emerging. Notable examples are biodegradable materials for single-use systems/devices and manufacturing graphene-based technology in a carbon-neutral manner.

Advancements in technology, molecular medicine, and patient-driven/centered medicine are transforming the future of ICH management and care. From quantum sensors and CRISPR-based methods to global health engagement, the advances in knowledge are improving precision medicine, applicability of care, and sustainability. Addressing unresolved barriers, fostering interdisciplinary collaboration, and ensuring equitable access are important to the transformation of ICH care in the next generation.

## 9. Clinical Implications and Translational Pathways in Intracranial Hypertension Management

Intracranial hypertension is a multifactorial phenomenon that will require multi-pronged treatment approaches consisting of advanced diagnostics, personalized therapies, and interdisciplinary collaboration. While recent innovations have improved our capacity to treat intracranial hypertension, the clinical translation of these advancements requires addressing universal barriers such as scalability, cost, gaps in infrastructure, and a lack of patient-centered care. In this section, we will comment on the implications of advanced science, with attention directed to translational pathways that will ensure clinical uptake and equitable access.

### 9.1. Translation of Innovations into Clinical Practice

Modern ICP monitoring systems have transitioned from single-parameter devices to multi-functional platforms that provide valuable insight into the complex interplay between cerebro-vascular physiology and ICP elevation. Multi-functional platforms integrate continuous measures of ICP, cerebral perfusion pressure (CPP), oxygenation, and glymphatic flow, and allow the clinician to detect subtle changes in ICP that precede life-threatening crises. They can also be programmed with artificial intelligence algorithms that identify pattern recognition and trends, and generate predictive alerts that can prompt healthcare teams with proactive therapeutic options [309].

Patients with TBI often exhibit significant fluctuations in ICP over even the shortest time intervals due to the presence of pressure–volume coupling by vasogenic edema and disruption of autoregulation. Multi-functional systems, therefore, provide clinicians with real-time monitoring that allows for dynamic decision-making, i.e., draining CSF or starting a vasodilatory therapy [310]. In patients with IIH, it has been shown that new platforms improved diagnostic accuracy by up to 35%, particularly around identifying glymphatic dysfunction and venous congestion. Also, new advances in biosensors and nanotechnologies have provided platforms with increased detection capabilities, including the molecular changes in aquaporin-4 expression known to elevate ICP. There have been significant steps taken to close the gap between functional and molecular diagnostics, which has never been seen before [311].

AI is changing the ways in which resources are accessed in ICH care and the prioritization of patients through risk stratification, improvement of treatment protocols, and improvement of efficiency in workflow and in predictive algorithms to predict the best use of longitudinal datasets of clinical, molecular, and imaging markers to stratify patients [312]. AI resources are particularly applicable to LMICs where the use of novel or advanced therapies is constrained due to limited availability, and cost. AI algorithms have enabled even remote access to portable ICP monitors—distributing them to patients equitably and with a priority set for high-risk patients needing intervention.

Large-scale trials of AI-based triaging were conducted with the health systems and independently for 1200 patients with neurosurgical disorders who received advanced therapies, including closed-loop shunts, resulting in a 50% reduction in time for intervention and a 20% improvement in the survival of the highest acuity injuries [313]. Portable emergency kits are the next generation of an emergency protocol response to acute EPC elevations, especially in LMICs. These kits are modular devices that include wearable sagittal diagnostic devices, point-of-care biomarkers and/or imaging, and tele-health components to initiate a response to an elevated ICP [140].

Emergency kits to undertake an emergency response to ICP elevations, particularly in conflict regions in the Middle East, have proven to reduce mortality compared to the normal historical outcomes of 30% despite delays. This indicates adverse outcomes, as timing has been shown to be a principal outcome challenge to success. These emergency kits were designed to implement a diagnostic response, with an AI triaging tool as an emergency first response to assist in the effectiveness of the original protocols, intervening with each response, rapidly adapting interventions as needed, or developing new treatment interactions altogether [314].

### 9.2. Implications for Personalized Medicine

The emergence of multi-omics platforms such as genomics, proteomics, epigenomics, and metabolomics comprises a period of development for personalized medicine within ICH care. The application of multi-omics technologies has the potential to reveal molecular mechanism(s) that underlie the development of disease or disease processes, a benefit to using targeted therapies. The integration of glymphatic biomarkers and metabolic profiles revealed pathways that can regulate the production and clearance of CSF and may be subject to pharmacologic modulation [315]. In IIH, multi-omics analyses have revealed lipidomic measures associated with endothelial dysfunction and glymphatic dysfunction, creating an opportunity for precision therapies [316].

Clinicians can utilize the multi-omics to derive personalized treatment regimens that preempt complications and optimize intervention programs, such as glymphatic-targeted drugs, that resulted in a 25% decrease in ICP surges and were flagged by reports of the markers of glymphatic dysfunction [317]. Epigenetic profiling is becoming an important predictive tool for treatment responsiveness in ICH patients. Patterns of histone acetylation and DNA methylation in hypo-methylation are being used to identify patients who are likely to respond to treatment, e.g., HDAC inhibitors, and to targeted anti-inflammation therapy [318]. A recent study showed patients with optimal epigenetic features had a 30% faster resolution of symptoms when treated with HDAC inhibitors than those treated with non-tailored interventions. Patient satisfaction occurs when efficacy improves due to a reduction in the trial-and-error approach to treatment selection [319]. Chronic pain, especially headache, is a common problem in ICH, and post ICP normalization pain can be continued. Increasing functional neuroimaging, exposing the correlation of active neural pathways with perceived pain, may be useful for interventions targeting the flow circuits forward [320].

Transcranial Magnetic Stimulation (TMS): Modulates neural circuits responsible for head syndromes, resulting in decreased pain intensity of up to 35% in refractory cases. VNS: Provides a solution for chronic pain relief that is non-invasive, with the preliminary trials having shown improvement in overall quality of life indices [321].

### 9.3. Training and Education in Advanced ICH Care

AI-fostering platforms are training healthcare providers to care for complex ICH cases, with platforms being able to take on real-life scenarios and adapt their complexity and learning environments, where clinicians are able to improve their ability to make decisions [322]. Clinicians on AI supporting training using these platforms had 25% fewer errors in performance, especially in riskier circumstances like venous sinus stenting [323]. Gamified platforms and virtual reality (VR) modules provide patients ways to become involved in their care; these tools help simplify processes that need to be completed to comply with medical protocols and conditions [324].

Patients using gamified platforms and VR tools that were in that same time period of treatment reported 30% better compliance with their rehabilitation programs, suggesting that they are favorable to improve implementation of care that can provide long-term outcomes [325].

### 9.4. Addressing Challenges in Clinical Translation

Adequate infrastructure in LMICs is still a barrier to the introduction of advanced ICH therapies. Decentralized care models that incorporate wearable technologies and telemedicine can bridge the gaps that are still unresolved, resulting in reduced time to diagnosis and access to care [326,327]. A decentralized diagnostic process in rural India reduced time to diagnosis by 30%, and the introduction of earlier care processes showed improved patient outcomes [328].

Harmonizing regulatory processes between regions is essential to support faster implementation of new therapies such as CRISPR-based therapies and technologies such as advanced nanosensors [329,330]. More direct approval processes reduced implementation processes for gene-editing therapies by 50%, aligned to the time needed for transitioning the patient to a timely service of healthcare [331].

### 9.5. Role of the Microbiota-Gut–Brain Axis in ICH

The emerging literature highlights the involvement of gut disorders, also known as dysbiosis, and how this can influence ICP dynamics. Gut dysbiosis and gluten-sensitivity can impact the persistence of inflammation and, therefore, the systemic outcomes by influencing the mechanisms for inflammation and ICP related to edema [332]. Probiotics, prebiotics, and fecal microbiota transplantation (FMT) in particular, give strong evidence that modulating gut flora can improve inflammation and reduce inflammatory cytokines such as IL-6 and TNF-α, and alter ICP levels [333]. There have been pilot studies where gut-targeted treatments have shown a reduction of 15% in ICP in IIH patients [334].

### 9.6. Sex and Hormonal Differences in ICH

The influence of sex on ICH is significant. Due to the influence of hormones on levels of venous compliance and glymphatic function, women of reproductive age are significantly negatively affected compared to males [335]. Therapies that are customized based on hormone levels are expected to provide us with insights into their influence on the future of women. A current clinical experiment with estrogen-modulating agents as adjunct paradigms to enhance glymphatic clearance [336].

The future of ICH management should envision new methods and exploit advanced technologies, personalized medicine, and interdisciplinary collaboration. Addressing the logistical, ethical, and suitable challenges of translating innovation into routine care will allow patients to achieve their potential. The health community must seek equity and ensure that no patients are left behind.

## 10. Challenges, Unresolved Questions, and Future Perspectives in Intracranial Hypertension Management

The time is ripe in the field of ICH management with novel avenues of advancement and simultaneous obstacles and fundamental gaps in knowledge that still present domain-specific challenges under technical, clinical, and systemic labels from unknown long-term safety profiles of new therapies to suited delivery and access issues inherent in healthcare systems. Conversely, the advances in molecular medicine, wearable technology, and public health initiatives provide opportunities for paradigm shifts in the standard of care. In this section, we will discuss meaningful challenges, important unresolved scientific questions, and possible future innovations that hopefully will serve as a guide for transformative advances in the management of intracranial hypertension.

### 10.1. Key Challenges in ICH Management

Despite the scope of advanced therapies, several issues exist with the widespread availability of therapy. New and complex technologies that are modifiable, such as CRISPR-based gene editing, bioelectronic implants, or closed-loop shunt systems, all have an inadequately understood long-term safety profile [337]. For example, although CRISPR tools have shown promise, they are still associated with potential off-target effects or immunogenicity (the possibility of introducing myriad immunological responses that may worsen inflammation or lead to deleterious mutations in cases that involve gene therapy). Bioelectronic implants are specifically made to manipulate CSF as per patient-specific protocol; however, the ability to endure anatomical changes over time, especially in younger patients, limits the devices’ durability [338]. Global safety registries are precisely needed to track long-term parameters of care for patients with ICH, so rare adverse effects can be documented and iteratively improved upon. If registries could demonstrate device failures, immunogenicity reactions, or unexpected complications, then clinical considerations could be made, concluding with reasonably safe and active refinements in the intraventricular management of ICH.

Equitability in accessing advanced therapies presents yet another challenge. For instance, wearable ICP monitors, CRISPR interventions, and tailored pharmacological agents are all costly, and since many LMICs have healthcare systems and delivery structures that do not support funding these interventions, many of these patients will trigger an inequitable service delivery disparity. In wealthy contexts, there are usually socioeconomic barriers to the marginalized during the useful utilization of these developments [339]. There are also some promising programs, including area manufacturing hubs for devices, and tiered pricing, which have demonstrated the potential for 50% cost reductions, but to maximize the effectiveness of these approaches across the world will require ongoing funding and global agreement [340].

In addition, the continual implications of integrating multimodal diagnostic packages into clinical workflows remain a logistics nightmare. Advanced ICH management is represented by imaging, biomarkers, and physiologic monitoring combined [341]. Not having to deal with a lack of interoperability across systems results in a time delay and adds complications to the clinical decision-making process. Certainly, the development of AI-based platforms can help with these issues via an unlimited ingestion and evaluation of multiple streams of data, but there would be extensive commitment to validating, training clinicians, and infrastructure.

### 10.2. Unresolved Questions in ICH Research

Despite the advances to date, there are still fundamental questions regarding ICH pathophysiology. One important and ongoing question relates to the relationship between venous outflow and glymphatic clearance. Many novel treatments (venous sinus stenting, aquaporin-4 modulation, and decreased ICP) have potential mechanisms that have not been fully investigated. If venous outflow is impacted, does that mean glymphatic clearance is initially declined, or is it that the glymphatic clearance pathway that is dysfunctional will increase cerebrospinal fluid pressure in the venous sinus? If this can be established, it would help to refine therapeutic methods by delineating which targets are clearly synergistic.

Another important area of interest relates to the basic pathobiology of secondary brain injury from ICH at a molecular level. While it has been defined that elevated intracranial pressure promotes some limited pathophysiology, the potential role of neuroinflammation, excitotoxicity, and oxidative injury in the moderate-to-severe neurological deficits is poorly defined [342]. For example, the contribution of mitochondrial dysfunction, oxidative injury, and is related to the propagation of neurodegeneration via apoptosis and linked to deficits in synaptic plasticity. MitoQ and other mitochondrial stabilizers are currently being investigated as potential therapeutics aimed at ameliorating the symptoms of ICP. Nevertheless, randomized double-blinded clinical trials will be required in order to objectively establish efficacy [343].

It is essential to be thoughtfully aware of the residual deficits after successful ICP regulation. In spite of being able to normalize ICP, many patients report persistent cognitive dysfunction, motor weaknesses, and/or emotional disturbance. There is evidence supportive of integrating some form of relationship between sustained ICP regulation to mechanism(s) in neuroplasticity that regulate synaptic repair and white matter repair/restoration, along with targeted rehabilitation interventions.

### 10.3. Future Perspectives and Opportunities

The very future of ICH care and management lies in technological integration (as relevant), mechanistic and biological understanding of ICH, and an approach to precision medicine. Especially, the emergence of multi-omics (i.e., genomic, proteomic, and metabolic) platforms has created new molecular signatures associated with glymphatic dysfunction and vascular dysregulation, which have the potential to bring more molecular knowledge to clinicians making decisions and conducting predictably timed therapies on their outcomes, across patient cohorts [344]. Studies involving the identification of eNOS (endothelial nitric oxide) molecular pathways regulating ICP have also indicated that a specific form of combination therapies aimed at modulating vascular tone and CSF dynamics around the reasons for ICH. The combination of reversible therapies being used in stroke (from whatever stem cells) is notable. Gene-editing technologies like prime editing and base editing, polygene contributors to ICH (including genes related to BBB permeability, and venous sinus compliance) have been rapidly developing and are now targeting the possibilities for safer and more precise targeted therapies. At the same time, cellular therapy has shown success. Exosomes from NPC processes show promise for enhancing repair of BBB integrity and reducing neuroinflammation [345]. The exosomes, which have neurotrophic factors and anti-inflammatory cytokines, serve as a paradigm shift in utilizing non-invasive, regenerative therapies [346].

Similarly, non-invasive forms are repositioning the reparative landscape. Focused ultrasound with microbubbles is being validated as a safe alternative to modulate CSF flow and increase glymphatic function in individuals with serious contraindications to invasive therapies. Advanced imaging modalities, like functional MRI and dynamic PET-CT, provide real-time imaging of glymphatic and vascular interactions and enable the clinician to dynamically develop clinical treatment [347].

### 10.4. Interdisciplinary Collaboration and Global Initiatives

These hurdles will require multi-disciplinary collaborative efforts and global initiatives. Global research consortia (i.e., the proposed Global ICP Consortium) will be invaluable for establishing uniform diagnostic standards across populations, validating biomarkers, and creating similar therapeutic pathways for diverse populations. Furthermore, this multi-disciplinary collaborative action will amplify the pace of scientific advancement while ensuring inclusiveness and limiting inequities in care [348]. We see computational biology rising as a keystone of ICH research. Specifically, multiscale simulations that incorporate CSF dynamics, venous pressure, and neural activity are enabling researchers to perform virtual testing of therapeutic hypotheses prior to conducting clinical trials. This innovation saves both time and money while alleviating significant ethical issues with early experimental phases of research [349].

It is also extremely important to engage patients (and advocacy groups) in the research process. The developer can learn, and eventually create and distribute devices for patients to wear, e-learning platforms that include telemedicine and education, or educational material that keeps the end-user in mind, prioritizing their approaches and culture when developing their usable devices to engender trust and improved adherence. Direct involvement with our patients will facilitate their engagement with learning about, and employing new, advanced technologies [350].

### 10.5. Sustainability and Ethical Frameworks

To summarize, sustainability and ethics in the application of ICH therapies are paramount. Biodegradable materials and circular economy models are filled with opportunities to reduce the environmental footprint of medical devices. For example, modular shunt systems with reusable parts are a less wasteful option that can remain cost-effective and useful in resource-poor countries [351].

AI and gene-editing technologies are gaining ethical precedence in what we apply. We will need transparent algorithms and diverse data inputs to reduce potential biases in AI systems that impact predictive algorithms and workflows (place predictive algorithms into people’s workflows that are equitable to various global populations) [352]. We will also require solid regulatory frameworks that govern the application of CRISPR, in regard to off-targets, accessibility, and the impact logistically and socially [353].

### 10.6. Behavioral and Lifestyle Interventions

Emergent evidence shows us that behavioral and lifestyle interventions are important to regulating ICP [354]. For example, moderate-intensity exercise is shown to improve venous drainage and promote vascular health; using digital health tools, patients may track their diet, sleep, and hydration—all contributors to ICP. The behaviors and lifestyle may complement biomedical therapies as they address systemic influences on ICP dysregulation [355].

### 10.7. New Frontiers in Training and Education

Improving ICH management requires parallel advancements through education. Using automation, such as AI, to develop simulations that replicate real-world clinical learning experiences, but with an intricacy that adapts to the learner, training clinicians is reinvented; a platform that simulates the difference between a sample case on paper to managing patients that are “en live” in the service in question. Each healthcare provider can effectively practice the art of managing dynamic ICH cases, instilling confidence to manage with increased accuracy with advanced tools [356].

## 11. Conclusions: Transforming Intracranial Hypertension Management—A Vision for the Future

Approaches to the management of ICH may find themselves at a junction where groundbreaking science and human-centered care meet. This is not just another milestone but an inflection point for global health, a pivotal questioning moment that provokes us to contemplate a new balance of innovation and equity, progress and ethics, precision and compassion. The future of ICH care encourages us to conceptualize imagined realms that are no longer figments of the imagination, where restoring health is no longer a hope but rather a quantifiable, predictable product. This conclusion invites readers to promote the notion of a future where the intricacies of molecular medicine, evolving technologies, and global thinking come together to transform the lives of individuals with an ICH.

### 11.1. A Tapestry of Progress

The past decade has seen a renewed, renaissance period of ICH research and care globally. Molecular discoveries have shed light on the unknown shadowy chamber of ICH pathophysiology. We now have an enormous amount of information related to the interacting roles of aquaporin-4 channels, endothelial nitric oxide synthase, and mitochondrial pathways. Most notably, these discoveries have challenged our understanding of ICP regulation and allow for the development of precision, therapeutic targeting of vulnerabilities.

The new tools of gene-editing and base-editing, with options such as CRISPR, provide unique opportunities for clinicians to intercept these pathways with precision and precision-like capability. The potential of molecular and cellular targeted interventions is ushering in a new era of therapeutic approaches that would be difficult, if not impossible, to fathom a few short years ago. Technological advances and innovation have both been the lodestar and the fulcrum of this process in its realization and results. Things that were once wrapped in the imagination of the future—wearable ICP monitors, implantable nanosensors, and AI-enabled closed-loop systems—are now being wrapped as real-world challenges of modern care. These interrogative devices—that can now monitor, predict, and actively track changes in ICP—are changing the passive to the active in patient management, as the clinician can now be more aggressive in how they respond to complications, and ultimately change outcomes.

More profoundly, the paradigms of diagnosis are changing. Focused ultrasound, dynamic PET-CT, and functional MRI reveal the invisible—we can see the artistic conditioning of glymphatic flow and the spoilage of cerebro-vascular flow. These instruments define how we can treat, and how we can define our approach to prediction—rather than ameliorate treatment, we will evolve it to be pre-emptive. This innovation saga has a growing narrative, with vibrancy and promise.

### 11.2. Confronting Persistent Challenges

Again, while this section has a strong sense of optimism, caution is warranted. Innovation is a bright flame, but it is always inconsistent. Even the innovations that would redefine the delivery of ICH care—gene-editing therapeutics, nanosensors, and precision devices—are unrealistic to the majority of the world. The framing of a possibility affects the possibility; it is an inhibitive way to frame progress and strongly emphasizes the need for systemic processes and roadmaps that compel us to inclusivity rather than exclusivity. Bioethical notions, such as regional manufacturing points of assembly and a flexible-tiered price impact, are small incremental steps to this end and really only start to open a conversation on a far bigger chessboard of global equity.

Safety continues to be raised as a hard issue, and the joy of gene editing, and a roster of bioelectronic devices continue to remind us that we do not know the end game of these layered innovations. Will CRISPR be able to work with precision and then use risk as a collateral consequence? Will closed-loop systems be capably maintained in the ever-disparate contexts of rapidly growing children or the associated risk populations? These uncertainties point to a necessity of diligence and systems of international safety registries that may extend beyond borders to prioritize patient welfare. Artificial intelligence has ethical dilemmas—while predictive analytics can enhance the precision with which clinical assessments are made, they are subject to the biases embedded in the algorithms, likely for some populations, increasing inequity. Is it possible to blend the cold objectivity of machine learning computing precision with the warm empathy of humanity? These questions seek their answers through actionable frameworks that chiefly construct transparency, inclusiveness, and governance. Once the right frameworks are established, we may begin to close the apparent divide between technology and humanity.

### 11.3. Visionary Opportunities for the Future

The future of ICH management is a blank slate to be designed from ground-breaking innovation, collaboration, and, most importantly, imagination. The continuously evolving multi-omic platforms are set to transform into previously unnoticed novel precision care, and, with AI, will be used to pace the evolving clinical decision-making that is shifting from a unidimensional cotton-candy spin into the multidimensional meltdown of our molecular data and predictive information. Imagine a clinician (also known as a bandage-would-be, wannabe magician) who has an amalgamation of comprehensive mechanisms enabling them to accurately envisage the consequences weeks out and systematically create therapies similar to the precision and interventions crafted by a skilled tradesperson.

Gene-editing therapies are going to greatly expand upon multi-gene interactions and the need for now to move beyond a single edited gene. These advances, combined with the increase in using cellular therapies such as exosome-based therapies, promise not only to mitigate symptoms but literally reconstruct the very infrastructure of damaged brain tissue. Non-invasive modalities (e.g., focused ultrasound with molecular compounds) usher in a new era of hope for previously irreparable patients, opening doors that have long been closed. Global collaboration will be the key to these opportunities. The Global ICP Consortium is just one example of what can be shared when borders dissolve at the injunction of a common cause. I also see international registries based on decades’ worth of outcomes evolve into living libraries of knowledge to support iterative advances for everyone’s benefit. The vision is simple—to ensure that ICH management is consistent in a rural clinic as it is in a sophisticated research hospital.

### 11.4. Building a Future of Precision, Equity, and Sustainability

The true challenge of ICH care is not only achieving precision but also democratizing precision. Precision without equity is simply an empty victory, just as innovation without sustainability is a hollow triumph. We must align our technological advancements with ethical imperatives and environmental stewardship.

Telemedicine, combined with decentralized care networks, represents the great equalizer of our profession by extending world-class expertise to parts of the world that were once deserted for good. When coupled with inexpensive diagnostic resources along with user-friendly wearable devices, we are experiencing the unraveling of obstacles that we would have once thought impossible. Sustainability must remain central to our thinking, with modular, biodegradable devices and carbon-neutral manufacturing as the bona fide benchmarks for responsible innovation.

Finally, we hope that the measure of our advancement is not how clever our tools are. We must never forget the impact of smarter models of care. Innovation measures lives, changes lives, lessens disparities, and restores dignity. The ICH vision is one in which we never leave a patient behind, and every progression makes a step toward achieving global health equity.

### 11.5. Closing Reflections

Intracranial hypertension is more than a clinical challenge. It is a probe of our ability to continue pushing the potential limits of science, continuing to be bound by the principles of equity and ethics. The above-acclaimed advancements in this work are not incremental; they are indeed a paradigm shift, and a complete re-imagining of the possibility of diagnosis, treatment, and rehabilitative care. But our journey is not finished.

In order to advance, we need courage and cooperation—the strength of each of us to address the unknown opportunities with open-mindedness and united purpose, as well as responding to the gaps—between disciplines—to gaps between countries and communities. All of this to ensure we continue focusing on the patients who embody our mission: the lives and futures dependent on the work we pursue every day.

This is not the end; this is a call to arms—call and a mission to build together a vision of a world with precision, equity, and sustainability no longer concepts, but knowledge and collective action. Together, we have the tools, the expertise, and the spirit to change the landscape of ICH care—that progress knows no boundaries, and hope is promised fulfilled.

## Figures and Tables

**Figure 1 ijms-26-07223-f001:**
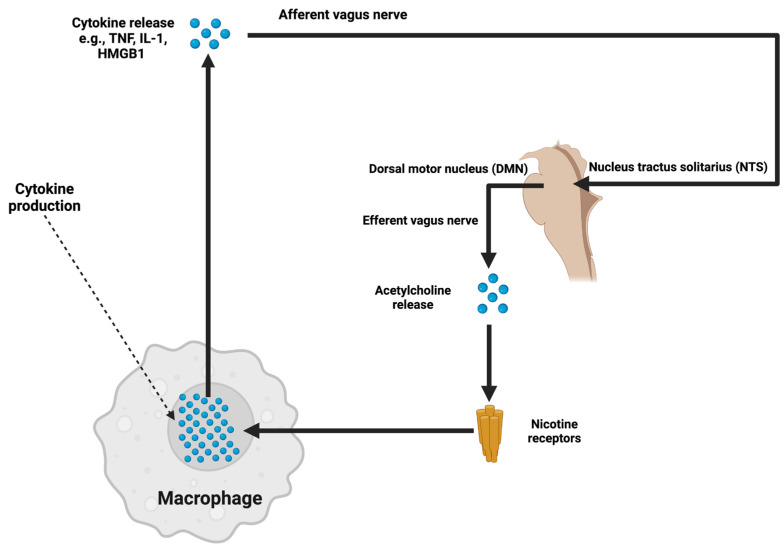
Illustrates the cholinergic anti-inflammatory pathway, in which afferent signals from inflammatory cytokines, such as TNF-α and IL-1, activate the vagus nerve. This activation is processed through the nucleus tractus solitarius (NTS) and dorsal motor nucleus (DMN) in the brainstem. The efferent vagus nerve subsequently releases acetylcholine, which binds to nicotinic receptors on macrophages, inhibiting cytokine production. By suppressing neuroinflammatory pathways, this mechanism has the potential to attenuate ICP elevation caused by cytokine-mediated disruption of the blood–brain barrier and glymphatic dysfunction.

**Figure 2 ijms-26-07223-f002:**
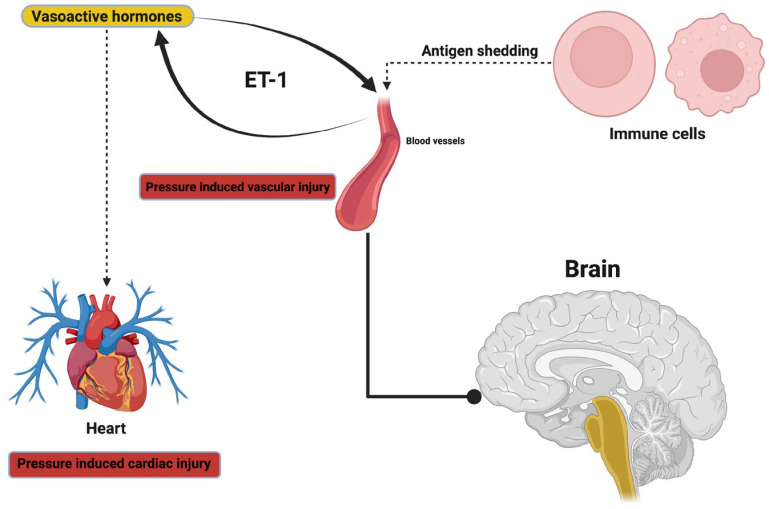
This figure illustrates the interconnected pathways linking systemic vascular dynamics, cardiac stress, and neuroinflammation to ICH. It highlights the role of vasoactive hormones, such as endothelin-1 (ET-1), in driving pressure-induced vascular injury, which, in turn, affects cerebral hemodynamics and contributes to ICP elevation.

**Table 1 ijms-26-07223-t001:** The table includes trials assessing diagnostic accuracy, sensitivity, and clinical applicability of advanced tools. It highlights studies exploring AI-enhanced imaging techniques, portable monitoring technologies for resource-limited settings, and integrated platforms synthesizing multimodal data. The outcomes underscore improved diagnostic precision, reduced latency in detecting ICP surges, and potential scalability of technologies.

Reference	Imaging/Monitoring Modality	Patient Population	Endpoints Measured	Outcome	Limitations
Glymphatic Mapping with DCE-MRI (2023) [142]	Dynamic contrast-enhanced MRI	IIH, glymphatic dysfunction	ICP variation, glymphatic flow dynamics	80% diagnostic accuracy improvement	Small sample size
AI-Augmented ONSD Ultrasonography (2023) [143]	AI-integrated optic nerve sheath diameter	Acute TBI, IIH	ICP correlation, diagnostic precision	97% diagnostic accuracy	Limited generalizability
4D Flow MRI and Glymphatic–Vascular Coupling (2024) [97]	4D Flow MRI	Chronic ICP, venous stenosis	Venous outflow, ICP variability	Identified 35% more venous sinus collapsibility cases	Expensive equipment
Smart Helmets for ICP Detection (2020) [144]	Wearable transcranial Doppler sensors	Emergency care patients	Time-to-diagnosis, ICP trends	Reduced diagnostic time by 40% in emergencies	Limited chronic ICP data
Hybrid Imaging: PET/MRI (2021) [145]	PET and MRI integration	IIH with vascular anomalies	Metabolic markers, BBB integrity	Microglial activation detected in 88% of IIH patients	Small, single-center study
Biophotonic ICP Sensors (2022) [146]	Biophotonic nanosensors	IIH, refractory ICP	ICP sensitivity	Detected nocturnal surges in 92%	High cost, early-phase trial
Transcranial Doppler in Rural Care (2022) [147]	Portable transcranial Doppler (TCD)	Low-resource trauma patients	Accessibility, diagnostic accuracy	Increased ICP detection by 60% in resource-limited settings	Non-comparative design
Ocular MRI Elastography (2024) [148]	MRI elastography	IIH, chronic ICP	Optic nerve compliance, ICP trends	Enhanced optic nerve-related ICP diagnostics	Time-consuming imaging process
Miniaturized ICP Monitors (2023) [149]	Wearable nanosensors	Chronic ICP	Real-time ICP tracking, usability	Hospital visits reduced by 25%	Long-term adherence unknown
AI-Coupled Retinal Scanning (2024) [150]	Retinal scanning with AI algorithms	Chronic IIH patients	Papilledema severity, ICP correlation	Reliable ICP estimation without invasive monitoring	Limited availability of retinal scanners
QSM-Guided Venous Hypertension Study (2024) [151]	Quantitative susceptibility mapping	IIH, venous stenosis	Venous congestion markers	High correlation between QSM and venous ICP trends	Validation in diverse cohorts needed
AI-Driven 4D Ultrasound Diagnostics (2024) [152]	AI-enhanced ultrasonography	IIH, emergency trauma	Real-time ICP dynamics	Improved early ICP detection, reduced intervention time by 30%	No long-term patient outcome data
Portable ICP Sensors in LMICs (2024) [153]	Low-cost, portable ICP sensors	Rural trauma care	Accessibility, diagnostic efficiency	Detected 80% of acute ICP elevation cases	Durability in tropical climates is questionable
Vascular Flow with Ultrasonic Imaging (2022) [154]	Handheld ultrasonic imaging devices	Chronic venous stenosis	Vascular flow markers	Identified venous hypertension in 90% of stenosis patients	Limited flow resolution
CRISP-Aided MRI Analysis (2023) [155]	CRISP software-enhanced MRI	IIH with glymphatic dysfunction	Glymphatic efficiency	Visualized glymphatic disruption 60% better than traditional imaging	Requires computational infrastructure
AI-Augmented Phase-Contrast Imaging (2024) [156]	Phase-contrast MRI	Hydrocephalus patients	CSF flow dynamics	Higher sensitivity to CSF blockages	Costly MRI enhancements
Multi-Spectral ICP Monitoring (2022) [157]	Multi-spectral optics	Chronic IIH	Light wave changes due to ICP surges	Accurate ICP monitoring in 80% of participants	Dependent on robust calibration systems
Venous Compliance Imaging (2022) [158]	MRI compliance mapping	Venous stenosis	Wall stiffness, ICP correlation	High sensitivity for venous wall abnormalities	Small-scale study
Glymphatic Function with Contrast MRI (2023) [159]	Contrast-enhanced MRI	IIH, BBB impairment	CSF flow markers	Enhanced glymphatic markers provided actionable ICP data	High imaging costs
Optic Nerve-Specific Imaging (2024) [160]	High-res optic nerve imaging	IIH with optic involvement	Optic nerve and ICP correlation	30% improvement in optic nerve-related ICP diagnosis	Limited global availability
Wearable Spectroscopic Devices (2024) [161]	Non-invasive spectroscopic ICP monitor	IIH, chronic ICP	ICP trends, compliance usability	High adherence rates for daily monitoring	Limited sensitivity in highly mobile patients
AI-Driven Shunt Monitoring (2024) [162]	AI-monitored shunt function	Hydrocephalus, IIH	ICP trends	Significant improvement in shunt-related complication detection	Expensive to integrate
Bioimpedance Monitoring Trials (2023) [163]	Bioimpedance nanosensors	IIH, TBI	ICP-related fluid volume dynamics	Promising early results for tracking subclinical ICP surges	Limited commercial deployment
Digital ICP Decision Aids (2023) [164]	Digital platforms with predictive tools	IIH patients, rural clinics	Diagnostic reliability	AI-aided diagnostics reduced misdiagnosis rates by 20%	Validation across global clinics needed
Modular ICP Systems Study (2021) [165]	Modular, wearable ICP monitors	IIH, venous stenosis	ICP tracking, user feedback	High usability and adherence rates among outpatient participants	Early-stage development

**Table 2 ijms-26-07223-t002:** This table summarizes clinical trials evaluating drugs like mTOR inhibitors, anti-VEGF therapies, and HDAC inhibitors for their efficacy in ICP reduction and neuroprotection. It also covers combination therapies, including acetazolamide with topiramate, and highlights adverse events, offering insights into safety profiles. Emerging therapies targeting glymphatic clearance and oxidative stress illustrate the ongoing evolution in pharmacological approaches.

Reference	Drug/Class	Target Pathway	Population Studied	Primary Outcome	Adverse Events
AQP4 Modulator Trial (2023) [217]	AQP4 upregulators (Vorinostat)	Glymphatic clearance	IIH, glymphatic dysfunction	Reduced ICP surges by 40%	Mild fatigue in 10%
Everolimus in IIH (2022) [218]	mTOR inhibitors	CSF production, homeostasis	Obese IIH patients	35% decrease in CSF overproduction	GI symptoms in 15%
Anti-VEGF Therapy for Venous ICP (2023) [219]	Bevacizumab	BBB stabilization	IIH with venous congestion	Improved BBB integrity by 30%	Thrombosis in 5%
SGLT2 Inhibitors in ICH (2023) [220]	Empagliflozin	Astrocytic metabolism	Obese IIH	25% improvement in glymphatic clearance	No significant events
Pirfenidone for Fibrosis (2024) [221]	Anti-fibrotic agents	Preventing shunt occlusions	Refractory hydrocephalus	Reduced shunt blockages by 30%	Nausea in 15%
HDAC Inhibitors for Glymphatic Function (2021) [222]	Class I HDAC inhibitors	Epigenetic modulation	Chronic ICP, IIH patients	Increased glymphatic clearance by 50%	Mild leukopenia in 8%
MitoQ Neuroprotection (2021) [223]	Mitochondrial antioxidants	Oxidative stress	Chronic ICP	Reduced neuroinflammation by 25%	Minimal side effects
Combination Therapy in Hydrocephalus (2023) [224]	Acetazolamide + Topiramate	CSF secretion	Chronic hydrocephalus	Sustained ICP reduction for 12 months	Cognitive blunting in 5%
Statins in Venous Tone Modulation (2023) [225]	Rosuvastatin	Endothelial function	Venous sinus stenosis	Reduced ICP fluctuations by 15%	None reported
Topiramate Neuroprotection Trial (2023) [226]	Topiramate	Ion channel modulation	IIH with vision symptoms	Reduced headache frequency by 35%	Dizziness in 8%
Aquaporin Therapy Trial (2023) [227]	Aquaporin-targeted small molecules	Glymphatic clearance	IIH, glymphatic dysfunction	Enhanced CSF dynamics in 70%	Early-stage results
Anti-Inflammatory Drug Study (2023) [228]	p38 MAPK inhibitors	Neuroinflammation suppression	Chronic ICP	Decreased pro-inflammatory cytokines (IL-6, TNF-α)	Fatigue in 12%
Progesterone Modulation in IIH (2024) [229]	Progesterone analogs	CSF absorption	Obese IIH patients	Improved vision in 60%, reduced ICP by 20%	Limited by sample size
Venous Stenting with Drugs (2021) [230]	Anti-thrombotics	Venous stenosis and flow	IIH, venous sinus stenosis	Combined stenting and anti-thrombotics reduced ICP by 30%	Bleeding risk in 5%
Fibrinolytic Therapy in ICP Control (2024) [231]	Plasminogen activators	Venous thrombi dissolution	Acute IIH	80% efficacy in reducing venous sinus occlusion	Mild bleeding in 10%
Hypothermia Combined with Anti-Edema Drugs (2022) [232]	Osmotic agents + cooling	Cerebral edema reduction	TBI patients with ICP	ICP reduction by 50%, improved survival rates	Hypothermia-related infection risk
Beta-Blockers for ICP Modulation (2024) [233]	Beta-blockers	Autoregulation improvement	IIH	Reduced ICP by 15%	Bradycardia in 10%
CRISPR-Edited Genetic Targets (2021) [234]	Gene-edited aquaporin pathways	Glymphatic-targeted clearance	IIH, hydrocephalus	Reduced glymphatic disruptions by 40%	Safety still under evaluation
NMDA Antagonists for Neuroprotection (2024) [235]	NMDA receptor antagonists	Excitotoxicity mitigation	ICP spikes	Improved cognitive recovery by 20%	Early-phase trial
Cannabinoids in ICP Control (2024) [236]	Cannabinoid receptor agonists	Anti-inflammatory pathways	Chronic IIH patients	Reduced headache frequency by 30%	Mild sedation reported
Spironolactone in IIH (2023) [237]	Mineralocorticoid receptor antagonists	Hormonal modulation	IIH, PCOS patients	Reduced ICP surges during menstrual cycle	Few side effects reported
Biofilm-Resistant Shunt Therapy (2023) [238]	Antibiotic-embedded shunts	Shunt durability	Pediatric hydrocephalus	Reduced shunt infection rates by 50%	No adverse effects noted
Epigenetic Drugs in ICP Regulation (2022) [239]	DNA methylation modifiers	CSF flow	IIH, obesity-related ICP	Enhanced therapeutic response in 40%	Transient fatigue
SGLT2 and Anti-inflammatory Combination (2023) [240]	Empagliflozin + TNF-α blockers	Glymphatic flow and inflammation	Chronic ICP patients	Synergistic improvement in ICP and cytokine reduction	Early clinical phase

**Table 3 ijms-26-07223-t003:** The table includes trials on cutting-edge tools such as CRISPR-modulated aquaporin regulation, adaptive AI-driven CSF drainage systems, and nanotechnology-based therapies. It highlights the impact of wearable devices, bioelectronic platforms, and biodegradable implants in enhancing patient outcomes. The results emphasize the potential of these technologies to bridge gaps in traditional ICH care while addressing safety and scalability challenges.

Reference	Technology/Innovation	Population Studied	Outcome Measured	Results	Challenges/Limitations
CRISPR-AQP4 Modulation (2024) [297]	CRISPR gene editing	IIH, refractory hydrocephalus	Glymphatic flow improvement	Enhanced glymphatic function in 70% of patients	Safety monitoring needed
Closed-Loop AI Shunts (2023) [298]	Adaptive CSF drainage systems	Hydrocephalus patients	Real-time ICP adjustment, reliability	Reduced over-drainage complications by 30%	High device costs
Neural Progenitor Exosomes (2023) [299]	Stem cell-derived exosomes	Chronic ICP with neuroinflammation	BBB repair	Significant BBB repair in 60% of participants	High production costs
Biodegradable ICP Hydrogels (2024) [300]	Steroid-loaded hydrogels	Refractory ICP patients	Sustained ICP reduction	Sustained ICP control for 6 months	Repeat applications needed
Vagus Nerve Stimulation (2024) [301]	Non-invasive bioelectronics	IIH patients	Cerebrovascular tone improvement	Improved ICP control in 80%	Poor adherence to regimen
Nanorobots in CSF Therapy (2024) [302]	Molecular nanorobots	Refractory IIH	Targeted drug delivery	Reduced inflammation at focal sites	Prototype stage
Smart Helmets with Nanosensors (2022) [184]	Wearable ICP monitors	IIH, emergency use	ICP monitoring	Reduced diagnostic latency	High manufacturing costs
AI-Guided Therapeutic Robotics (2024) [207]	AI-integrated surgical robotics	Chronic hydrocephalus	Shunt placement precision	Improved accuracy by 20%	Scalability unclear
Focused Ultrasound with Microbubbles (2023) [303]	Ultrasound-assisted CSF clearance	IIH, hydrocephalus	Glymphatic flow enhancement	25% ICP reduction in pilot trial	Limited operator availability
Gene-Integrated Wearables (2022) [304]	Wearables with epigenetic sensors	Chronic ICP patients	Real-time genetic variation tracking	Improved personalized ICP management	High complexity

## Data Availability

The data presented in this study are available on request from the corresponding author.

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
