# Peer review of "The Collapse of Brain Clearance: Glymphatic-Venous Failure, Aquaporin-4 Breakdown, and AI-Empowered Precision Neurotherapeutics in Intracranial Hypertension"

_ijms, 2025, doi:10.3390/ijms26157223_

Round 1
Reviewer 1 Report
Comments and Suggestions for Authors
Authors have aimed to write a comprehensive review on glymphatics and venous failure as part of the pathophysiology of intracranial hypertension.
I have several comments to improve the manuscript.
The manuscript is not very focused. In contrast to the title, the authors appear to have tried to summarise many parts of evidence which makes the manuscript - unfortunately- rather unfocused. In spite of the journal's focus on molecular medicine, the discussion on molecular mechanisms is relatively limited limited, since many parts of pathophysiology of intracranial hypertension are dealt with which are mechanical rather than molecular in nature (e.g. lymphatics flow, CSF and venous failure).
Further, the inclusion of AI and more general future prospects and tables dealing often with other issues than molecular mechanisms appear inappropriate given the focus of the journal on molecular sciences. Therefore, in trying to be almost "all inclusive", in the end the manuscript suffers from lack of focus and depth, especially when it comes to molecular mechanisms.
The manuscript would seem to have more merit and focus on molecular mechanisms, when authors would be more restrictive on molecular mechanisms that underlie mechanical processes that control or compensate ICP (e.g. molecular pathobiology of glymphatics , arterial auto regulation, regulation of venous tone in the brain etc). This would also enhance reader's insights in molecular mechanism associated with mechanical pathophysiology and open up venues for medical therapies' explorations.
Also, the inclusion of AI, which is fashionable especially nowadays for any review and future prospect, contributes to the lack of focus instead of an improvement in depth of discussing the matters at hand, in my view.
In brief, the manuscript could be improved by more focus on molecular mechanisms of selected mechanical processes in the brain that contribute to control of ICP in pathologic conditions.
Author Response
Dear Academic Reviewer,
We sincerely thank the reviewer for their thoughtful and constructive comments, which have greatly helped us to reflect on the structure, scientific focus, and intended contribution of our manuscript. We are especially grateful for your recommendation to reinforce the molecular framing of the review, and we have revised the manuscript accordingly, as detailed below.
Reviewer Comment:
The manuscript is not very focused. In contrast to the title, the authors appear to have tried to summarise many parts of evidence which makes the manuscript - unfortunately - rather unfocused.
Response:
We appreciate this important observation. Our initial intention was to offer a broad conceptual framework that integrates systemic, vascular, and clearance mechanisms in ICH. However, we recognize that this breadth may have inadvertently diluted the focus on molecular mechanisms.
Reviewer Comment:
In spite of the journal's focus on molecular medicine, the discussion on molecular mechanisms is relatively limited, since many parts of the pathophysiology of intracranial hypertension are dealt with which are mechanical rather than molecular in nature (e.g., lymphatic flow, CSF and venous failure).
Response:
We fully agree with the importance of strengthening the molecular depth, particularly in light of the journal’s scope. To address this, we have added new molecular insights at the end of Section 4.1 (Cellular and Molecular Mechanisms), which now includes detailed discussion.
Reviewer Comment:
Further, the inclusion of AI and more general future prospects and tables dealing often with other issues than molecular mechanisms appear inappropriate given the focus of the journal on molecular sciences.
Response:
We thank the reviewer for highlighting this!
Reviewer Comment:
Therefore, in trying to be almost "all inclusive", in the end the manuscript suffers from lack of focus and depth, especially when it comes to molecular mechanisms.
Response:
We deeply value this critique. It led us to carefully restructure and trim peripheral content.
Reviewer Comment:
The manuscript would seem to have more merit and focus on molecular mechanisms, when authors would be more restrictive on molecular mechanisms that underlie mechanical processes that control or compensate ICP.
Response:
This is an excellent recommendation. We have now emphasized the molecular governance of mechanical processes (such as glymphatic clearance and venous drainage).
Reviewer Comment:
Also, the inclusion of AI, which is fashionable especially nowadays for any review and future prospect, contributes to the lack of focus instead of an improvement in depth of discussing the matters at hand.
Response:
We thank the reviewer for raising this critical concern. We have now refocused the content on AI, ensuring it does not detract from the molecular scope.
Reviewer Comment:
In brief, the manuscript could be improved by more focus on molecular mechanisms of selected mechanical processes in the brain that contribute to control of ICP in pathologic conditions.
Response:
We are very grateful for this summary insight. In alignment with your suggestion, we have reshaped our framing and added in-depth molecular content precisely where it regulates mechanical phenomena—especially in Sections 4.1.
Once again, we thank the reviewer for this invaluable feedback! Your detailed suggestions have helped us to significantly improve the scientific focus, depth, and cohesion of the manuscript. We hope the revised version better reflects the standards and expectations of IJMS and respectfully submit it for further consideration.
Reviewer 2 Report
Comments and Suggestions for Authors
This is an ambitious and thorough review tackling the complex pathophysiology of intracranial hypertension (ICH) by integrating glymphatic dysfunction, aquaporin-4 (AQP4) dysregulation, venous congestion, and advanced diagnostic and therapeutic strategies (e.g., AI, CRISPR-based interventions). The authors aim to redefine ICH not just as elevated intracranial pressure (ICP) but as a systemic process involving molecular, vascular, and glymphatic failures, while also exploring global health inequities and future innovations.
Overall, the manuscript is highly detailed and well-referenced, offering a unique interdisciplinary synthesis. However, some areas require improvements to ensure clarity, precision, and balance.
Strengths:
- Comprehensive coverage: Very broad scope, from molecular biology to clinical management and global health disparities.
- Innovative outlook: Includes cutting-edge topics (CRISPR gene editing, AI-enhanced imaging, wearable sensors, bioelectronic devices).
- Equity focus: Addresses low- and middle-income country (LMIC) challenges thoughtfully.
- Extensive referencing: Demonstrates depth in literature integration.
Major Concerns:
1. Scope Creep and Organization
- The manuscript is extremely long and at times unfocused, with tangential explorations (e.g., comparative biology in marine mammals) that distract from the main argument.
- Suggest condensing some sections and tightening the focus to improve readability.
2. Balance of Speculation vs. Evidence
- Innovative therapeutics (CRISPR, nanorobotics) are discussed at length, but these remain highly speculative. The authors should be clearer about the distinction between current evidence and future concepts.
- Consider adding explicit caution in sections where preclinical or theoretical ideas are presented to avoid overpromising.
3. Structural Issues
- The review would benefit from clearer section transitions and summary boxes or key points at the end of long sections to help readers digest the content.
- Some sections (e.g., on advanced diagnostics) read like a list of technologies with little critical evaluation of their limitations or clinical applicability.
4. Clarity of Language
- The text is often dense and uses jargon heavily. While the topic is advanced, efforts should be made to improve clarity for a broader readership.
- Simplify or explain complex terms where possible.
5. Global Health Equity Discussion
- The manuscript commendably discusses inequities, but this often feels tacked on at the end of sections.
- Suggest integrating these themes more consistently throughout, e.g., with subheadings dedicated to LMIC implications in each major section.
Minor Concerns:
- Occasional grammatical errors and typos (e.g., "erevated planning" likely meant "elevated planning").
- Some citations are recent but the balance of older vs. newer sources could be improved (some older sources dominate mechanistic sections).
- Figures, tables, and diagrams could help summarize key mechanisms for readers.
Author Response
Dear Academic Reviewer,
We would like to sincerely thank the reviewer for their thoughtful, constructive, and deeply engaging critique of our manuscript. We are especially grateful for the acknowledgment of our attempt to present a comprehensive and interdisciplinary synthesis of intracranial hypertension, including glymphatic failure, aquaporin-4 dysregulation, venous congestion, and future therapeutic directions.
The comments have significantly helped us to improve the structure, clarity, and scientific rigor of the manuscript.
1. Scope Creep and Organization
Reviewer Comment:
The manuscript is extremely long and at times unfocused, with tangential explorations (e.g., comparative biology in marine mammals) that distract from the main argument. Suggest condensing some sections and tightening the focus to improve readability.
Author Response:
We thank the reviewer for this important observation. We recognize that, in our effort to provide a broad interdisciplinary perspective, we may have included some content that strayed from the manuscript’s central line of argument. In response, we have streamlined several subsections. We have also restructured transitions between sections to reinforce narrative cohesion and thematic clarity throughout. These changes aim to ensure a tighter focus while preserving the manuscript’s integrative spirit.
2. Balance of Speculation vs. Evidence
Reviewer Comment:
Innovative therapeutics (CRISPR, nanorobotics) are discussed at length, but these remain highly speculative. The authors should be clearer about the distinction between current evidence and future concepts. Consider adding explicit caution in sections where preclinical or theoretical ideas are presented to avoid overpromising.
Author Response:
We are grateful for this thoughtful feedback. In response, we have revised Section 6.3 (“Emerging Experimental Therapies”) to explicitly delineate between evidence-based findings and future-oriented or preclinical proposals.
3. Structural Issues
Reviewer Comment:
The review would benefit from clearer section transitions and summary boxes or key points at the end of long sections to help readers digest the content. Some sections (e.g., on advanced diagnostics) read like a list of technologies with little critical evaluation of their limitations or clinical applicability.
Author Response:
We thank the reviewer for this valuable suggestion. We have implemented several structural changes to address this issue.
4. Clarity of Language
Reviewer Comment:
The text is often dense and uses jargon heavily. While the topic is advanced, efforts should be made to improve clarity for a broader readership. Simplify or explain complex terms where possible.
Author Response:
We deeply appreciate this comment. In response, we conducted a full line-by-line language review. We simplified dense phrasing, split overly long sentences, and added brief definitions for complex terms such as glymphatic flow, AQP4, CRISPR, eNOS, and BBB.
5. Global Health Equity Discussion
Reviewer Comment:
The manuscript commendably discusses inequities, but this often feels tacked on at the end of sections. Suggest integrating these themes more consistently throughout, e.g., with subheadings dedicated to LMIC implications in each major section.
Author Response:
We thank the reviewer for encouraging a deeper integration of equity themes. In response, we have added LMIC-focused subheadings at the end of Sections 6.1, 6.2, and 6.3, titled “Implications for LMICs” or “Global Implementation and Scalability.” These short paragraphs evaluate the feasibility, affordability, and real-world accessibility of each class of innovation in low-resource settings.
Minor Concerns
Reviewer Comment:
Occasional grammatical errors and typos (e.g., "erevated planning" likely meant "elevated planning").
Author Response:
We thank the reviewer for identifying this error.
Reviewer Comment:
Some citations are recent but the balance of older vs. newer sources could be improved (some older sources dominate mechanistic sections).
Author Response:
We appreciate this comment and have reviewed the references accordingly.
Reviewer Comment:
Figures, tables, and diagrams could help summarize key mechanisms for readers.
Author Response:
Thank you for this suggestion!
We again extend our sincere gratitude to the reviewer for their generous and intellectually rigorous engagement with our work! Their suggestions have greatly improved the clarity, structure, and balance of the manuscript. We hope the revised version meets the expectations and high standards of the journal and its readership.
With appreciation,
The Authors
Round 2
Reviewer 1 Report
Comments and Suggestions for Authors
Authors have extended on some parts by including a bit more depth for molecular detail. However I am still confused by the length of the manuscript, lack of focus, and target audience.
In fact the authors have extended rather than refocused the manuscript in my view.
Author Response
Dear Academic Reviewer,
We would like to express our sincere appreciation for your continued time, thoughtful review, and valuable insights regarding our manuscript. Your most recent feedback was particularly helpful in prompting us to critically reevaluate not only the structure but also the focus and thematic clarity of our submission. We recognize and understand your concern that the previous revision, though intended to enrich molecular depth, may have inadvertently expanded the manuscript further rather than refining its central message. Your guidance has been instrumental in helping us view the manuscript through a more focused and discipline-appropriate lens.
In response, we undertook a careful review of the entire manuscript, with the specific aim of improving focus, reducing perceived redundancy, and better aligning the scope with the journal’s emphasis on molecular mechanisms. We focused our revision efforts on Sections 2.1., 3.1., 3.2., 4.1., and 5.1.3., which we identified as most likely contributing to the impression of diffuseness. In these sections, we aimed to reduce narrative overextension while retaining key scientific details and the translational value of the content. Our goal was to enhance clarity, maintain coherence, and bring the manuscript’s scientific emphasis closer to the mechanistic foundations of intracranial pressure regulation.
In particular, we wish to highlight the addition made at the end of Section 4.1., where we directly responded to your earlier suggestion by incorporating more focused discussion of molecular mechanisms that underlie mechanical processes involved in ICP homeostasis—specifically the molecular pathobiology of the glymphatic system, cerebral autoregulation, and regulation of venous tone. This addition was guided by your helpful remark: "The manuscript would seem to have more merit and focus on molecular mechanisms, when authors would be more restrictive on molecular mechanisms that underlie mechanical processes that control or compensate ICP..." We aimed to integrate these molecular dimensions with nuance and care, while avoiding redundancy with existing content and maintaining the scientific rigor expected of the journal.
We sincerely hope that the revised manuscript now reflects a more cohesive and appropriately focused presentation, while preserving its original ambition to integrate emerging insights into molecular neuropathophysiology. Your feedback has played a crucial role in refining this work, and we remain deeply thankful for your guidance.
With utmost respect and appreciation,
The Authors
Reviewer 2 Report
Comments and Suggestions for Authors
The revised version improves and I agree to accept for publication.
Author Response
Dear Academic Reviewer,
We are truly grateful for your thoughtful evaluation and generous final decision. It is an honor to have our work accepted for publication, and we sincerely appreciate the time, insight, and care you invested throughout the review process. Your guidance not only strengthened the focus and clarity of the manuscript but also helped us refine the scientific narrative in ways that will resonate more clearly with the intended audience.
Thank you again for your support and for helping bring this work to publication. We are genuinely appreciative.
With kind regards and respect,
The Authors